# Metabolic glycan labeling immobilizes dendritic cell membrane and enhances antitumor efficacy of dendritic cell vaccine

Joonsu Han [1], Rimsha Bhatta[1], Yusheng Liu[1], Yang Bo[1], Alberto Elosegui-Artola [2,3] & Hua Wang [1,4,5,6,7,8,9] ✉

Dendritic cell (DC) vaccine was among the first FDA-approved cancer immunotherapies, but has been limited by the modest cytotoxic T lymphocyte (CTL) response and therapeutic efficacy. Here we report a facile metabolic labeling approach that enables targeted modulation of adoptively transferred DCs for developing enhanced DC vaccines. We show that metabolic glycan labeling can reduce the membrane mobility of DCs, which activates DCs and improves the antigen presentation and subsequent T cell priming property of DCs. Metabolic glycan labeling itself can enhance the antitumor efficacy of DC vaccines. In addition, the cell-surface chemical tags (e.g., azido groups) introduced via metabolic glycan labeling also enable in vivo conjugation of cytokines onto adoptively transferred DCs, which further enhances CTL response and antitumor efficacy. Our DC labeling and targeting technology provides a strategy to improve the therapeutic efficacy of DC vaccines, with minimal interference upon the clinical manufacturing process.

Immunotherapy has shifted the paradigm for cancer treatment in the past decade, especially with the success of checkpoint blockades and chimeric antigen receptor (CAR) T cell therapy[1,2]. However, the low patient response rate towards checkpoint blockades, limited efficacy of CAR T cell therapy against solid tumors, and severe side effects for both have limited their utility[3–5]. The ability to modulate the function of antigen presenting cells (e.g., dendritic cells (DCs)) and orchestrate the priming of antigen-specific T cells holds promise to improve the overall cytotoxic T lymphocyte (CTL) response and therapeutic efficacy, and has been actively pursued in therapeutic cancer vaccines[6,7]. Indeed, Sipuleucel-T, one type of DC vaccines for treating prostate cancer, was among the first FDA-approved cancer immunotherapies[8]. While various other types of cancer vaccines have been actively pursued[9–12], Sipuleucel-T remains the only FDA-approved therapeutic vaccine despite a modest therapeutic benefit (4.1-month increase in

median survival)[13]. Strategies to improve the antitumor efficacy of cancer vaccines, especially DC vaccines, while maintaining the benign safety profile are greatly demanded.

DC vaccines function by isolating DCs from the patient's blood, expanding them in the presence of tumor antigens and adjuvants, and infusing back to patients[13]. Extensive efforts have been made to optimize the source of DCs and antigens, the protocol for expanding antigen-presenting DCs ex vivo, and the administration routes of DCs[14–17]. For example, to address the low number of DCs in peripheral blood, bone marrow-derived DCs (BMDCs) were explored as the DC source[18–20]. The cell culture conditions and cytokine cocktails were screened for the improved proliferation of DCs ex vivo[21–23]. However, issues including suboptimal activation of DCs and lack of control over DC function after adoptive transfer remain largely unsolved. The potential dysfunction of adoptively transferred DCs during the blood

[1]Department of Materials Science and Engineering, University of Illinois at Urbana-Champaign, Urbana, IL 61801, USA. [2]Cell and Tissue Mechanobiology Laboratory, Francis Crick Institute, London, UK. [3]Department of Physics, King's College London, London, UK. [4]Cancer Center at Illinois (CCIL), Urbana, IL 61801, USA. [5]Department of Bioengineering, University of Illinois at Urbana-Champaign, Urbana, IL 61801, USA. [6]Carle College of Medicine, University of Illinois at Urbana-Champaign, Urbana, IL 61801, USA. [7]Beckman Institute for Advanced Science and Technology, University of Illinois at Urbana-Champaign, Urbana, IL 61801, USA. [8]Materials Research Laboratory, University of Illinois at Urbana-Champaign, Urbana, IL 61801, USA. [9]Institute for Genomic Biology, University of Illinois at Urbana-Champaign, Urbana, IL 61801, USA. ✉e-mail: huawang3@illinois.edu

circulation and tissue penetration, as a result of the shearing force, metabolic stress, and apoptotic signals, often undermines the resultant CTL response and therapeutic benefit, but strategies for in vivo targeted modulation of adoptively transferred DCs are lacking[24–26].

Here we introduce a metabolic labeling and targeting technology that enables targeted modulation of adoptively transferred DCs in vivo (Fig. 1a)[27–30]. We previously reported that DCs can be metabolically labeled with chemical tags (e.g., azido groups) via the metabolic glycoengineering process of unnatural sugars[31]. In this study, we carefully examine the impact of metabolic glycan labeling on the membrane property and activation status of DCs, and apply the metabolic glycan labeling and targeting technology to the context of DC vaccine. We find that the metabolic glycan labeling process itself is able to reduce the mobility of proteins on DC membrane and upregulate the activation markers of DCs (Fig. 1b), leading to the improved ability of DCs to process and present peptide and protein antigens and subsequently prime antigen-specific CD8+ T cells. By simply adding azido-sugars to the culture medium of DCs during the expansion of antigen-presenting DCs, an essential step for clinical manufacturing of DC vaccines, the CTL response and antitumor efficacy of DC vaccines are significantly improved. We further demonstrate that azido groups on the surface of DCs, introduced via metabolic glycan labeling, can mediate conjugation of dibenzocyclooctyne (DBCO)-modified agents (e.g., IL-15 and IL-2) via efficient and bioorthogonal click chemistry (Fig. 1a)[32–35]. The surface conjugation of IL-15[36,37] onto adoptively transferred, antigen-presenting DCs dramatically improves the T cell priming process, and further enhances the antigen-specific CD8+ T cell response and antitumor efficacy. This metabolic labeling and targeting technology provides a generalizable approach to modulating DCs and could potentially be used for the development of enhanced DC vaccines, with minimal interference upon the clinical manufacturing process.

## Results

### Metabolic glycan labeling activates DCs

Prior to the metabolic labeling study, we first synthesized tetraacetyl-N-azidoacetylmannosamine (Ac4ManNAz), a commonly-used small-molecule metabolic labeling agent (Fig. 2a)[32]. The metabolic glycoengineering process of Ac4ManNAz is expected to yield azido-tagged glycoproteins and glycolipids on the cell membrane. Ac4ManNAz was characterized via $^1$H and $^{13}$C NMR spectrometry (Supplementary Fig. S1). By incubating BMDCs with Ac4ManNAz for 72 h and staining with DBCO-Cy5 for 20 min, a bright Cy5 signal was detected on the cell membrane (Fig. 2b), indicating the successful metabolic labeling of BMDCs with azido groups. Flow cytometry analysis confirmed the concentration-dependent metabolic labeling of BMDCs by Ac4ManNAz (Fig. 2c). Surprisingly, BMDCs treated with Ac4ManNAz showed significantly upregulated expression of CD86 and MHCII in comparison with untreated BMDCs (Fig. 2d, Supplementary Figs. S2a and S3a–d), indicating the enhanced activation status of DCs after treatment with azido-sugars. The CD86 expression level and percentage of CD86+MHCII+ BMDCs increased with the concentration of Ac4ManNAz (Fig. 2d, Supplementary Figs. S2a and S3a–d). Ac4ManNAz-treated DCs also expressed a higher number of CD40 and CCR7, additional activation markers of DCs, than untreated DCs (Fig. 2e, f, Supplementary Fig. S2b, c). To further understand whether the azido-labeled subpopulation showed increased expression of CD86 and MHCII than non-labeled subpopulation, BMDCs were treated with 0, 10, 50, and 200 μM Ac4ManNAz, respectively for 72 h and then stained with DBCO-Cy5 and FITC-conjugated anti-CD86 or FITC-conjugated anti-MHCII (Fig. 2g). Compared to the Cy5– subpopulation, Cy5+ (i.e., azide+) subpopulation showed a significantly higher expression level of CD86 and MHCII at all tested Ac4ManNAz concentrations (Fig. 2g–i, Supplementary Fig. S2d–f), substantiating the positive correlation between azido labeling and CD86/MHCII

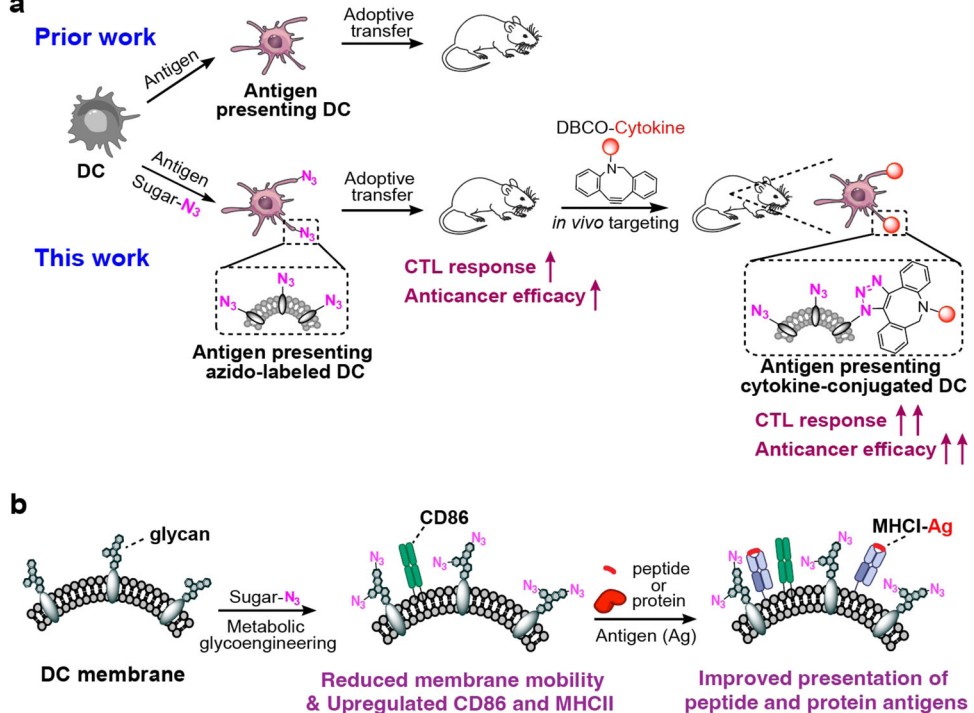

**Fig. 1 | Schematic illustrations of metabolic labeling and targeting of dendritic cells (DCs) for developing enhanced DC vaccines. a** Azido-sugars can be metabolized by DCs and become expressed on the cell membrane in the form of glycoproteins and glycolipids, which itself improves the activation status and antigen presentation ability of DCs. The azido-labeled DCs also enable conjugation of DBCO-cytokines (e.g., IL-15) post adoptive transfer, for further improved CTL response and antitumor efficacy. **b** We demonstrate that metabolic glycan labeling of DCs reduces the mobility of cell membranes, upregulate the surface expression of CD86 and MHCII, and improves the ability of DCs to process and present antigens.

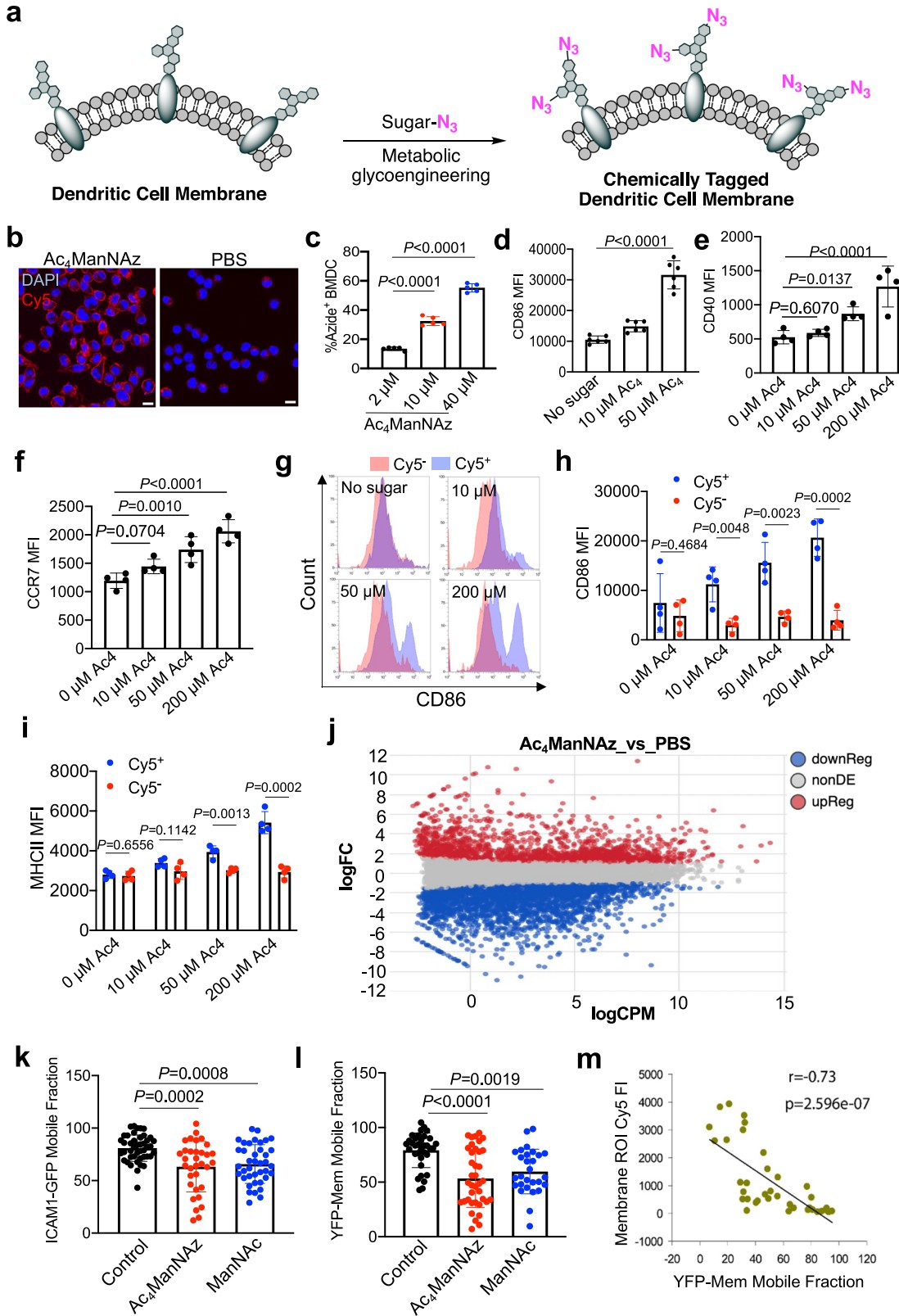

upregulation. It is noteworthy that Ac₄ManNAz treatment did not cause significant cytotoxicity towards BMDCs (Supplementary Fig. S2g), ruling out the impact of the potentially released danger signals by dying cells.

To understand whether the azido tag alone plays a role in DC stimulation, we compared the DC activation effect of Ac₄ManNAz and *N*-acetylmannosamine (ManNAc). Similar to Ac₄ManNAz, ManNAc was

also able to upregulate the surface expression of CD86 and MHCII on DCs in comparison with untreated DCs (Supplementary Fig. S3a–d). Compared to DCs treated with Ac₄ManNAz, DCs treated with ManNAc expressed similar levels of CD86 and MHCII (Supplementary Fig. S3a–d), ruling out the impact of azido tag alone on the activation of DCs. We also expanded our analysis to include other non-natural sugars such as tetraacetyl-*N*-azidoacetylgalactosamine (Ac₄GalNAz)

**Fig. 2 | Metabolic glycan labeling reduces DC membrane mobility and improves DC activation. a** Schematic illustration of metabolic labeling of DCs by Ac₄ManNAz and other azido-sugars. **b** CLSM images of BMDCs after 3-day treatment with Ac₄ManNAz and 20-min staining with DBCO-Cy5. Scale bar: 10 μm. This experiment was repeated at least twice with similar results. **c** Percentages of azide⁺ BMDCs after 3-day incubation with Ac₄ManNAz of varying concentrations, as detected by DBCO-Cy5 ($n = 5$). **d** CD86 mean fluorescence intensity of BMDCs after 3-day incubation with different concentrations of Ac₄ManNAz ($n = 6$). **e** CD40 mean fluorescence intensity of BMDCs after 3-day incubation with different concentrations of Ac₄ManNAz ($n = 4$). **f** CCR7 mean fluorescence intensity of BMDCs after 3-day incubation with different concentrations of Ac₄ManNAz ($n = 4$). **g** Representative CD86 histograms of DCs within Cy5⁺ and Cy5⁻ subpopulations after treated with different concentrations of Ac₄ManNAz for three days and stained with FITC-conjugated anti-mouse CD86 and DBCO-Cy5. Also shown are (**h**) mean CD86 fluorescence intensity of DCs within Cy5⁺ and Cy5⁻ subpopulations following the same treatment as in **g** ($n = 4$). **i** mean MHCII fluorescence intensity of DCs within Cy5⁺ and Cy5⁻ subpopulations ($n = 4$). **j** Altered gene expression profiles between Ac₄ManNAz-treated DCs and untreated DCs. DCs were treated with 200 μM Ac₄ManNAz or PBS for three days. **k** ICAM1 mobile fractions of DCs treated with Ac₄ManNAz, ManNAc, or PBS for 3 days, as determined by the FRAP assay of ICAM1-GFP expressing DCs ($n > 30$). **l** YFP-Mem mobile fraction of DCs treated with Ac₄ManNAz, ManNAc, or PBS for 3 days, as detected by the FRAP assay of YFP-Mem expressing DCs ($n > 30$). **m** Correlation between Cy5 fluorescence intensity of membrane regions of interest (ROIs) and YFP-Mem mobile fractions. YFP-Mem expressing DCs were treated with Ac₄ManNAz for 72 h and stained with DBCO-Cy5 for 30 min, prior to FRAP assays. All the numerical data in these figures are presented as mean ± SD (for **c–f**, **k**, and **l**, one-way ANOVA with post hoc Fisher's LSD test was used; for **h** and **i**, two-tailed Welch's $t$ test was used; $0.01 < *P \le 0.05$; $**P \le 0.01$; $***P \le 0.001$). Source data are provided as a Source Data file.

and tetraacetyl-*N*-azidoacetylglucosamine (Ac₄GluNAz). Both Ac₄GalNAz and Ac₄GluNAz treatment resulted in the upregulated expression of CD86 and MHCII on DCs, in comparison with untreated DCs (Supplementary Fig. S4a, b). To better comprehend the potential mechanism behind the upregulation of activation markers observed with metabolic glycan labeling, we also performed the transcriptome analysis of Ac₄ManNAz- and PBS-treated DCs. As a result, Ac₄ManNAz treatment resulted in widespread transcriptional changes of DCs, with 1947 genes upregulated and 2949 genes downregulated (Fig. 2j, Supplementary Figs. S5 and S6). The significantly upregulated genes include *H2-K2, H2-D1, IL-15, IL-1a, TNF-a* (Supplementary Fig. S6a–e), which is consistent with the improved antigen presentation and inflammatory phenotype of DCs. While CD86 did not show a significant change in mRNA levels, we did observe the upregulated expression of spic (*spi1/PU.1* related), a transcription factor regulating the translation of CD86 (Supplementary Fig. S6f, g).

**Metabolic glycan labeling reduces the mobility of DC membrane**

Considering that azido-sugars are eventually expressed on the cell membrane in the form of glycoproteins and glycolipids, which are known to play a critical role in defining the structure and properties of cell membranes, we wondered whether the metabolic labeling process results in the change of membrane properties such as membrane mobility, as a possible mechanism underlying the increased activation status of DCs. To demonstrate, we transfected DCs with the ICAM-1-GFP plasmid to express GFP-tagged ICAM-1 on the cell membrane. ICAM-1-GFP expressing DCs were treated with Ac₄ManNAz or ManNAc or PBS, followed by the fluorescence recovery after photobleaching (FRAP) assay to analyze the membrane mobility[38,39]. FRAP assays showed that DCs treated with Ac₄ManNAz or ManNAc exhibited a decreased mobile fraction of membrane ICAM-1, in comparison to control DCs without azido-sugar treatment (Fig. 2k, Supplementary Fig. S7a). When DCs were transfected with YFP-Mem (20 amino acids of neuromodulin) and analyzed by FRAP assays, a similar phenomenon, i.e., a decreased fraction of mobile YFP-Mem, was observed for DCs treated with Ac₄ManNAz or ManNAc in comparison to control DCs (Fig. 2l, Supplementary Fig. S7b). The overall recovery half-time of membrane ICAM-1 or YFP-Mem was negligibly changed with or without sugar treatment (Supplementary Fig. S7c, d). To further understand the correlation between azido labeling and membrane mobility for the Ac₄ManNAz group, we stained GFP-ICAM-1 or YFP-Mem expressing DCs, which were pretreated with Ac₄ManNAz for 72 h, with DBCO-Cy5 before FRAP assays. Membrane regions of interest with higher Cy5 signals showed lower mobile fractions of GFP-ICAM-1 or YFP-Mem (Fig. 2m, Supplementary Fig. S7e), substantiating the correlation between azido labeling and reduced membrane mobility. These experiments indicated that Ac₄ManNAz-mediated metabolic labeling processes reduced the mobility of cell-membrane proteins, which might have affected the receptor clustering and downstream

intracellular pathways that eventually contributed to the upregulation of CD86 levels. While the altered membrane fluidity could be related to the enhanced activation status of DCs, future efforts are needed to fully elucidate whether there is a causal relationship between them.

**Metabolically labeled DCs show improved processing and presentation of antigens**

In view of the enhanced activation status of DCs after treatment with azido-sugars, we next studied whether the metabolically labeled DCs show improved processing and presentation of antigens. We started with a model peptide antigen, SIINFEKL, considering the well-established characterization methods[40–42]. BMDCs were incubated with Ac₄ManNAz or PBS for three days, and incubated with SIINFEKL peptide (20 or 100 nM) for 16 h. At 100 nM SIINFEKL, a significantly increased level of MHCI-SIINFEKL complexes was detected on Ac₄ManNAz-treated DCs in comparison with untreated DCs (Fig. 3a, b), indicating the improved antigen presentation ability of azido-labeled DCs. Consistent with the activation results above, DCs treated with Ac₄ManNAz in the presence of SIINFEKL peptide expressed a significantly higher level of CD86 and MHCII in comparison with control DCs (Fig. 3c, d). To further validate the improved antigen presentation by azido-labeled DCs, we co-cultured the SIINFEKL-presenting DCs with carboxyfluorescein succinimidyl ester (CFSE)-stained OT-1 cells, which can specifically recognize MHCI-SIINFEKL complexes, for three days. As expected, Ac₄ManNAz-treated DCs resulted in a significantly improved proliferation of OT-1 cells, compared to DCs without azido-sugar treatment (Fig. 3e, f).

We next studied whether metabolically labeled DCs can better process and present protein antigens (e.g., ovalbumin (OVA)) than unlabeled DCs. BMDCs were incubated with Ac₄ManNAz or PBS for three days, and incubated with OVA protein (200 nM or 1 μM) for 16 h. At either OVA concentration, Ac₄ManNAz-treated DCs expressed a significantly higher level of MHCI-SIINFEKL complexes than control DCs (Fig. 3g). CD86 and MHCII expression levels were also upregulated in Ac₄ManNAz-treated DCs in comparison with untreated DCs in the presence of OVA (Fig. 3h, i). To further validate the improved antigen presentation by azido-labeled DCs, we incubated BMDCs with Ac₄ManNAz or PBS for three days and OVA protein (100 or 500 nM) for 16 h, and then cocultured them with CFSE-stained OT-1 cells. As a result, Ac₄ManNAz-treated DCs induced a significantly improved proliferation of OT-1 cells, in comparison with DCs without azido-sugar treatment (Fig. 3j, k). These experiments demonstrated that metabolic glycan labeling can improve the activation status of DCs, the processing and presentation of both peptide and protein antigens by DCs, and subsequent priming of antigen-specific CD8⁺ T cells in vitro.

**Metabolically labeled DCs show improved CTL response in vivo**

We next studied whether azido-labeled DCs pulsed with antigens would show improved antigen-specific CTL response in vivo in

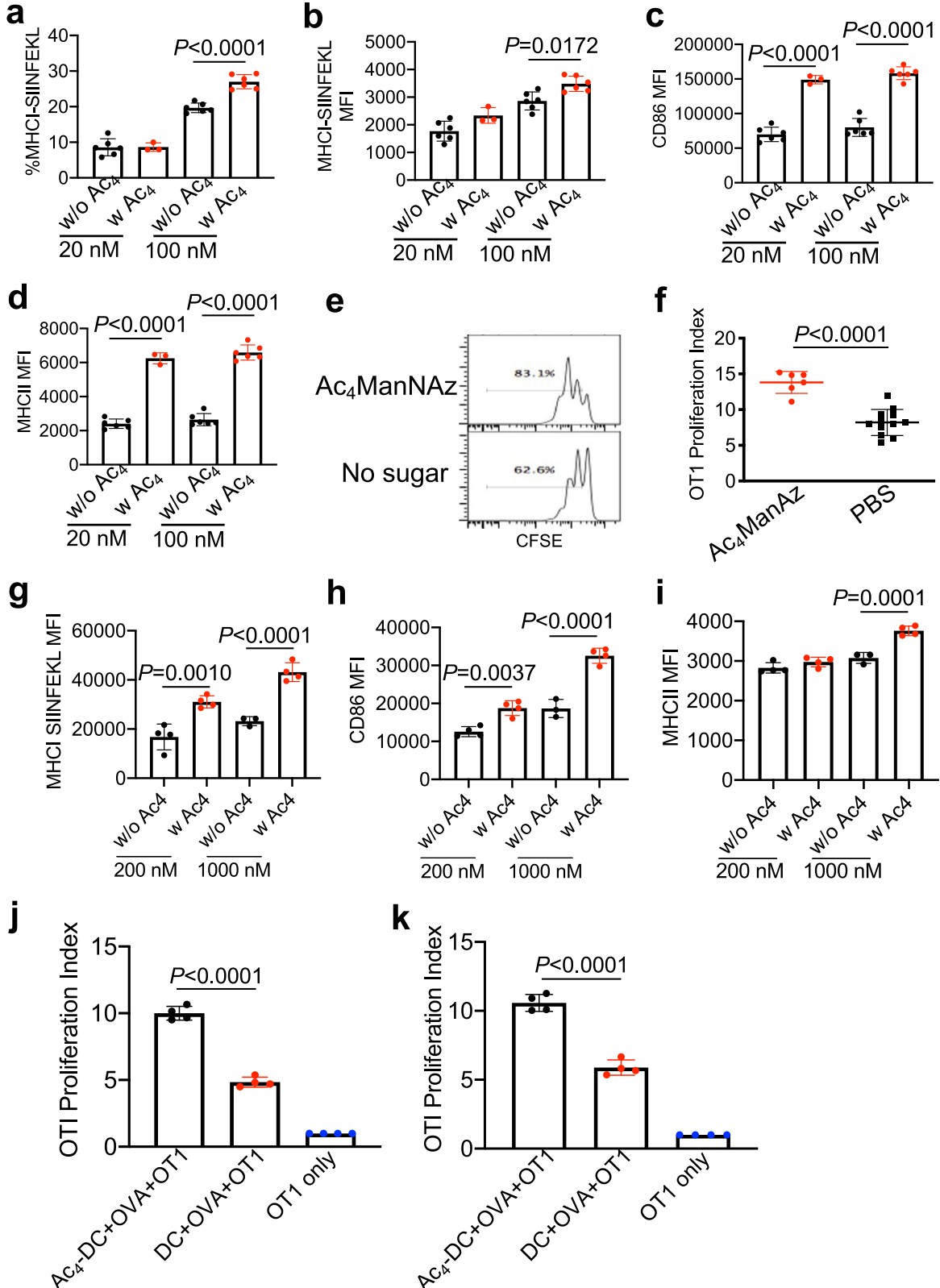

**Fig. 3 | Metabolic glycan labeling improves the activation status, antigen presentation ability, and subsequent T cell priming property of DCs. a–d** BMDCs were pretreated with Ac₄ManNAz (50 µM) or PBS for 3 days and incubated with SIINFEKL peptide (20 or 100 nM) for 16 h (*n* = 6). **a** Percentage of MHCI-SIINFEKL expressing BMDCs. Also shown are (**b**) MHCI-SIINFEKL, (**c**) CD86, and (**d**) MHCII expression levels of BMDCs. **e, f** Following the same treatment as in **a–d**, BMDCs were co-cultured with CFSE-stained OT-1 cells for 3 days. **e** Representative CFSE histograms of OT-1 cells. **f** Proliferation index of OT-1 cells after 3-day co-culture with Ac₄ManNAz- or PBS-pretreated DCs (*n* > 5). **g–i** BMDCs were pretreated with Ac₄ManNAz (50 µM) or PBS for 3 days and incubated with OVA protein (200 or 1000 nM) for 16 h (*n* = 6). Shown are (**g**) MHCI-SIINFEKL, (**h**) CD86, and (**i**) MHCII expression levels of BMDCs. **j, k** Proliferation index of OT-1 cells after 3-day co-culture with Ac₄ManNAz- or PBS-pretreated DCs (*n* = 4). DCs were pretreated with Ac₄ManNAz or PBS for 72 h and incubated with 100 nM (**j**) or 500 nM (**k**) OVA protein for 16 h, prior to coculture with OT-1 cells. All the numerical data in these figures are presented as mean ± SD (for **a–d, f–i**, two-tailed Welch's *t* test was used; for **j** and **k**, one-way ANOVA with post hoc Fisher's LSD test was used; 0.01 <*P* ≤ 0.05; **P* ≤ 0.01; ***P* ≤ 0.001). Source data are provided as a Source Data file.

comparison with unlabeled DCs (Fig. 4a). BMDCs were incubated with Ac₄ManNAz or PBS for three days and treated with SIINFEKL peptide for 16 h, prior to injection into C57BL/6 mice. At 10 or 20 days post injection of DC vaccines, peripheral blood mononuclear cells (PBMCs) were harvested for the analysis of SIINFEKL-specific CD8⁺ cells. Tetramer analysis showed an elevated level of SIINFEKL-specific CD8⁺ cells in both DC vaccine groups in comparison with the unvaccinated group (Fig. 4b, c). Compared to the unlabeled DCs, Ac₄ManNAz-treated DCs resulted in a higher frequency of SIINFEKL-specific CD8⁺ cells in PBMCs (Fig. 4b), indicating the improved ability of azido-labeled DCs to prime antigen-specific CD8⁺ T cells in vivo. Following PBMC analysis, E.G7-OVA tumors were subcutaneously inoculated in the flank of mice. Compared to the unvaccinated group, mice treated with DC vaccines showed significantly slower tumor growth (Fig. 4d, Supplementary Fig. S8a, b). Compared to control DCs without azido-sugar treatment, azido-labeled DCs resulted in slower tumor growth and prolonged animal survival (Fig. 4d, e, Supplementary Fig. S8a, b).

We also compared the antitumor efficacy of DC vaccines with or without Ac₄ManNAz pretreatment in a therapeutic setting (Fig. 4f). BMDCs were incubated with Ac₄ManNAz or PBS for three days and treated with SIINFEKL peptide for 16 h, prior to injection into C57BL/6 mice bearing established E.G7-OVA tumors. A single dose of DC vaccine on day 7 did not result in a significant therapeutic benefit compared to untreated mice (Supplementary Fig. S9a–d). By injecting DC vaccines on day 7 and 14, respectively, compared to the untreated group, both DC vaccine groups significantly reduced the growth rate of tumors and improved the survival of mice (Fig. 4g, h, Supplementary Fig. S9a–d). Compared to control DCs without azido-sugar pretreatment, Ac₄ManNAz-treated DCs further reduced the tumor growth rate and prolonged the survival of mice (Fig. 4g, h, Supplementary Fig. S10a–d). It is noteworthy that the tumors showed the sign of shrinkage after the second dose of DC vaccines but managed to grow back (Fig. 4g, Supplementary Fig. S10c, d). These experiments demonstrated that the simple treatment of DCs with azido-sugars

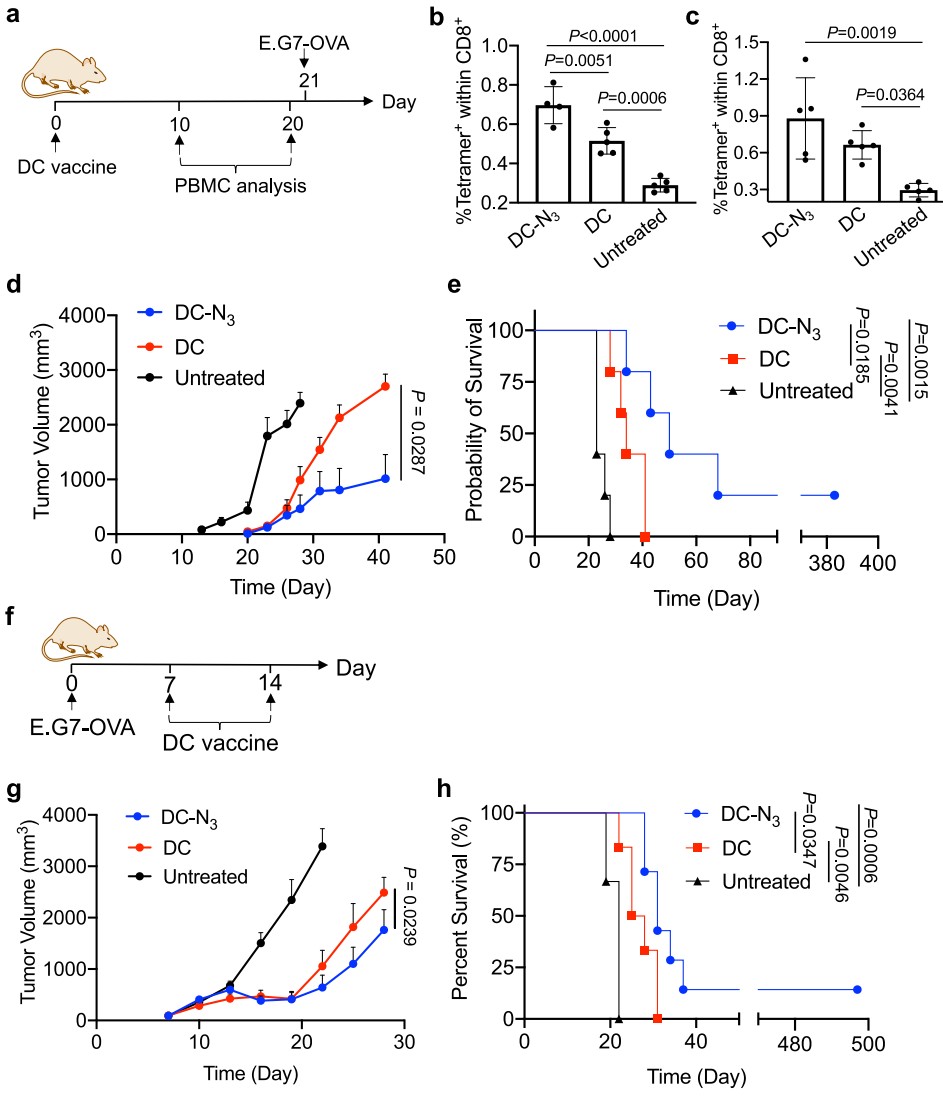

**Fig. 4 | Metabolic glycan labeling improves the CTL response and antitumor efficacy of DC vaccines. a** Timeframe of the vaccination study. DCs pretreated with Ac₄ManNAz for three days and pulsed with SIINFEKL peptide for 16 h were injected on day 0. E.G7-OVA tumor cells were subcutaneously inoculated on day 21. Shown are the percentage of SIINFEKL-specific CD8⁺ T cells in PBMC on (**b**) day 10 and (**c**) day 20, respectively (*n* = 5). **d** Average E.G7-OVA tumor volume of each group over the course of prophylactic tumor study. Noted that the date of tumor cell inoculation was counted as day 0 for the tumor volume curve (*n* = 5). **e** Kaplan–Meier plots for all groups (*n* = 5). **f** Timeframe of the tumor efficacy study. **g** Average E.G7-OVA tumor volume of each group over the course of therapeutic tumor study (*n* = 6-7). **h** Kaplan–Meier plots for all groups (*n* = 6-7). All the numerical data are presented as mean ± SD except for **d** and **g** where data are presented as mean ± SEM (for **b**, **c**, **e**, and **h**, one-way ANOVA with post hoc Fisher's LSD test was used; for d and g, two-tailed Welch's *t* test was used; 0.01 <*P ≤ 0.05; **P ≤ 0.01; ***P ≤ 0.001). Source data are provided as a Source Data file.

during the ex vivo manufacturing step can improve the CTL response and antitumor efficacy of DC vaccines.

## Azido-labeled DCs enable conjugation of cytokines for improved T cell priming

In addition to the DC-activating effect, azido labeling also enables targeted conjugation of DBCO-bearing immunomodulatory agents to DCs via efficient click chemistry, which can potentially further improve the antigen-specific CTL response and overall antitumor efficacy of DC vaccines (Fig. 5a). We first studied whether azido-labeled DCs can covalently capture DBCO-modified IL-2 and IL-15, two representative T cell-stimulating cytokines. DBCO-IL-2 (or DBCO-IL-15) were synthesized via the conjugation of amine-bearing IL-2 (or IL-15) and DBCO-sulfo-NHS, followed by purification via ultracentrifugation. Cy5-tagged DBCO-IL-2 and DBCO-IL-15 were also synthesized for quantification purposes. After treatment with Ac$_4$ManNAz or PBS for three days, DCs were further incubated with DBCO/Cy5-IL-2 or Cy5-IL-2 for 30 min. Compared to unlabeled DCs, azido-labeled DCs treated with DBCO/Cy5-IL2 showed higher Cy5 fluorescence intensity (Fig. 5b, c), indicating the successful conjugation of DBCO/Cy5-IL-2 to azido-labeled DCs via efficient click chemistry. Cy5-IL-2, instead, failed to show any targeting effect towards azido-labeled DCs (Fig. 5b, c). Similarly, DBCO/Cy5-IL-15 was able to conjugate onto azido-labeled DCs instead of unlabeled DCs (Fig. 5d, e). Cy5-IL-15 without DBCO modification failed to exert any targeting effect (Fig. 5d, e).

We next studied whether conjugation of IL-2 or IL-15 onto DCs can improve the priming of antigen-specific CD8$^+$ T cells. BMDCs were pretreated with Ac$_4$ManNAz for three days, incubated with DBCO-cytokines or unmodified cytokines for 30 min, and then cocultured with CFSE-stained OT-1 cells in the presence of SIINFEKL peptide for three days. Compared to the IL-15 group, azido-labeled DCs conjugated with DBCO-IL-15 resulted in significantly enhanced proliferation of OT-1 cells (Fig. 5f), indicating the improved stimulation of OT-1 cells by IL-15 conjugated onto the surface of DCs. By increasing the concentration of IL-15 by 5-fold, targeted conjugation of DBCO-IL-15 to azido-labeled DCs still resulted in improved proliferation of OT-1 cells than free IL-15 (Fig. S11a). Similarly, targeted conjugation of DBCO-IL-2 onto azido-labeled DCs also enhanced the proliferation of OT-1 cells in comparison with free IL-2, at varying IL-2 concentrations (Fig. 5g, Fig. S11b). It is noteworthy that the conjugation of IL-15 or IL-2 onto DCs resulted in comparable or higher expansion rates of OT-1 cells in comparison with the co-culture systems where IL-15 or IL-2 was present throughout the culture period (Fig. S11c–f), further demonstrating the superior T cell priming capability of IL-15/IL-2 conjugated DCs. To assess the potential off-target T cell activating effect of cytokine-conjugated DCs, we also co-cultured IL-15 conjugated DCs with both CFSE-stained OT-1 cells and Violet Stain-stained non-specific CD8$^+$ T cells for three days. As a result, OT-1 cells showed a significantly higher proliferation rate than non-specific CD8$^+$ T cells (Fig. 5h–j). Compared to non-specific CD8$^+$ T cells that were co-cultured with DCs, cells cocultured with IL-15-conjugated DCs failed to exhibit improved proliferation (Fig. 5h–j). To examine whether the surface conjugation of T cell-stimulating cytokines on DCs would induce the exhaustion of T cells, we analyzed the expression levels of inhibitory receptors (e.g., PD-1, CTLA-4, and LAG-3) on OT-1 cells. As a result, surface conjugation of DBCO-IL-15 or DBCO-IL-2 on DCs did not alter the surface expression of PD-1, CTLA-4, and LAG-3 by OT-1 cells after 3-day coculture (Fig. 5k–n, Fig. S11g, h).

## In vivo conjugation of IL-15 improves the antitumor efficacy of DC vaccines

We next studied whether DBCO-cargo can be conjugated onto azido-labeled DCs via click chemistry in vivo. It is noteworthy that azido groups expressed on the membrane of DCs can well retain for at least 72 h (Fig. S12), providing a durable time window for in vivo targeting of DBCO-bearing molecules. To examine the membrane retention time of

covalently conjugated molecules, BMDCs were pretreated with Ac$_4$ManNAz for three days and incubated with DBCO-biotin, followed by the detection of surface biotin groups via FITC-avidin at different times. It turns out that ~70% of covalently conjugated biotin could retain on DC membrane for 72 h (Fig. S13). To study the in vivo targeting of azido-tagged DCs, BMDCs pretreated with Ac$_4$ManNAz or PBS were stained with calcein AM and injected into C57BL/6 mice, followed by the injection of DBCO-Cy5 at 24 h. At 6 h post injection of DBCO-Cy5, lymph nodes were harvested for flow cytometry analysis (Fig. 6a, Fig. S14a, b). Compared to DCs without azido-sugar pretreatment, Ac$_4$ManNAz-pretreated DCs exhibited higher Cy5 fluorescence signal (Fig. 6b, c), indicating the successful conjugation of DBCO-Cy5 onto azido-labeled DCs in vivo. In contrast, endogenous DCs in the draining lymph nodes, i.e. calcein AM$^-$ DCs, showed minimal Cy5 signal (Fig. S14c). T cells, natural killer cells, and neutrophils in the draining lymph nodes also showed minimal Cy5 accumulation (Fig. S14c). These data demonstrated the specific conjugation of DBCO-Cy5 onto adoptively transferred, azido-tagged DCs. It is noteworthy that a similar amount of calcein AM$^+$ DCs were detected in the draining lymph nodes between Ac$_4$ManNAz and PBS groups (Fig. S14d), demonstrating the unaltered lymph node draining property of DCs after Ac$_4$ManNAz treatment. The adoptive transfer of Ac$_4$ManNAz-pretreated DCs or untreated DCs did not result in any noticeable changes in the number of immune cells such as natural killer cells in the draining lymph nodes (Fig. S14e). After demonstrating the in vivo DC targeting effect, we next explored whether in vivo conjugation of IL-15 to adoptively transferred, antigen-pulsed DCs can further improve the CTL response and antitumor efficacy. DC vaccines, prepared via treatment of BMDCs with Ac$_4$ManNAz for three days and SIINFEKL peptide for 16 h, were injected into C57BL/6 mice on day 0, followed by administration of DBCO-IL-15 or IL-15 on day 1, 2, and 3, respectively (Fig. 6d). After DC vaccination, a higher frequency of SIINFEKL-specific CD8$^+$ cells were detected in PBMC of mice treated with DBCO-IL-15 than mice treated with IL-15 (Fig. 6e, f), indicating that in vivo conjugation of DBCO-IL-15 to azido-labeled, SIINFEKL-expressing DCs can improve the expansion of SIINFEKL-specific CD8$^+$ T cells. In the following prophylactic tumor study, azido-labeled DC vaccine coupled with DBCO-IL-15 resulted in slower tumor growth and prolonged animal survival, in comparison with DC + IL-15 or DC alone (Fig. 6g, h).

We also examined the antitumor efficacy of azido-labeled DC vaccine + DBCO-IL-15 in a therapeutic setting. Following tumor inoculation on day 0, DC vaccines, prepared via treatment of BMDCs with Ac$_4$ManNAz for three days and SIINFEKL peptide for 16 h, were injected into C57BL/6 mice on day 6 and 9, respectively. DBCO-IL-15 or IL-15 or PBS was then administered on day 10, 11, and 12, respectively (Fig. 6i). Compared to the untreated group, all DC vaccine groups showed a reduced tumor growth rate and improved animal survival (Fig. 6j, k, Supplementary Fig. S15a, b). Compared to DC vaccine alone or the combination of DC vaccine and IL-15, DC vaccine coupled with DBCO-IL-15 further inhibited tumor growth and prolonged animal survival (Fig. 6j, k, Supplementary Fig. S15a, b). In contrast, DC vaccine coupled with IL-15 resulted in a negligible change in the tumor growth rate and animal survival compared to DC vaccine alone (Fig. 6j, k, Supplementary Fig. S15a, b). These experiments demonstrated that in vivo conjugation of IL-15 onto antigen-presenting DCs can improve antigen-specific CD8$^+$ T cell response and overall antitumor efficacy.

## Discussion

While metabolic glycan labeling has been widely used for chemical tagging of cells[27–35], to the best of our knowledge, its impact on the membrane property and function of cells such as DCs has not been reported. By transfecting DCs with ICAM1-GFP or YFP-Mem, we performed FRAP assays to compare the fluorescence recovery rate of

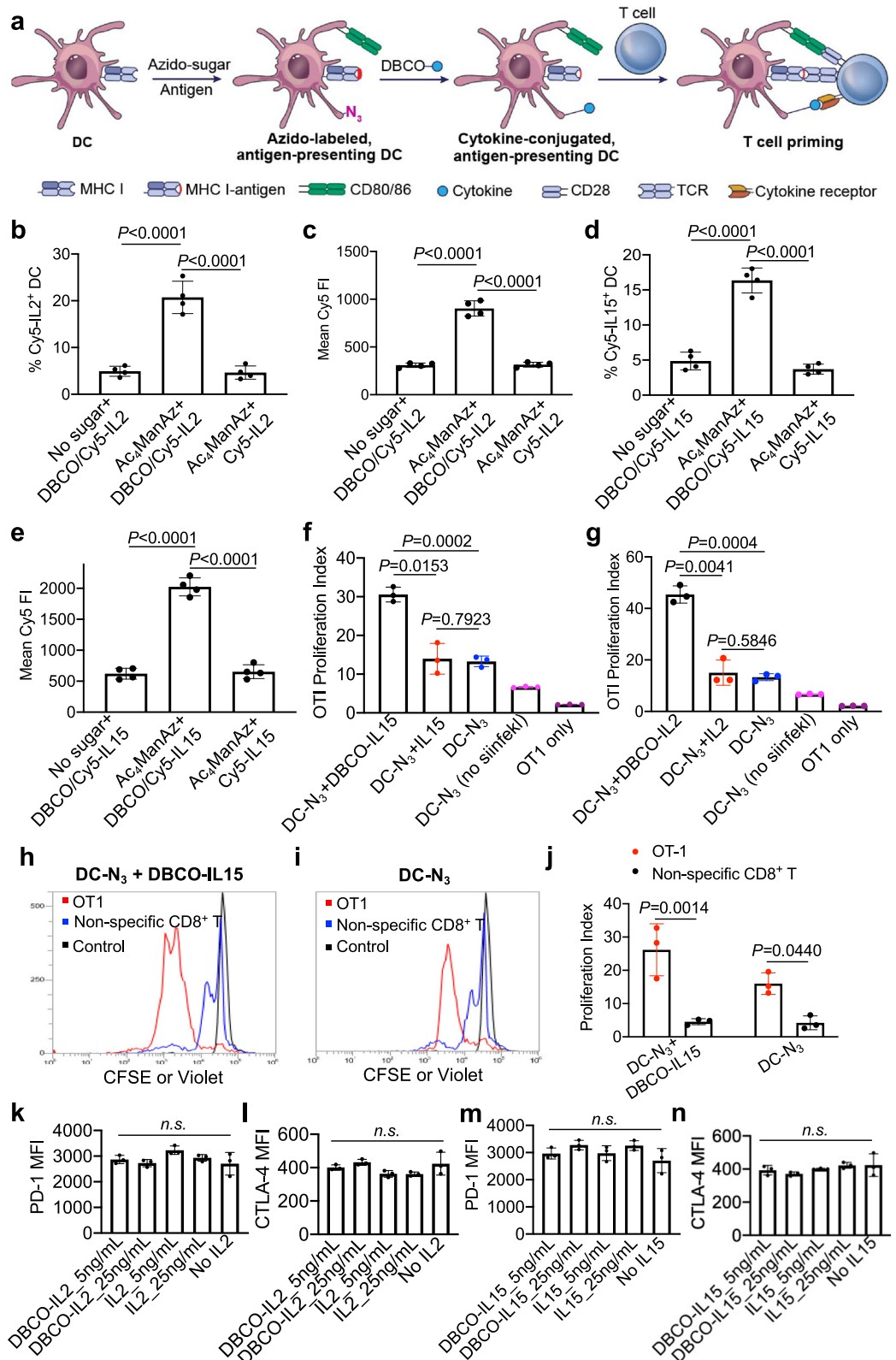

ICAM1-GFP or YFP-Mem, i.e. the mobility of membrane proteins. Compared to control DCs without azido-sugar treatment, DCs treated with azido-sugars exhibit a reduced membrane mobility (Fig. 2k−m). This can potentially explain why azido-labeled DCs show upregulated expression of activation markers (CD86, MHCII, CD40, and CCR7) (Figs. 2d−i and 3c, d, h, i), as stiffer membranes may contribute to receptor clustering and the activation of downstream signaling pathways[43−46]. We further showed that, compared to unlabeled DCs, azido-labeled DCs can better process and present both peptide and protein antigens (Fig. 3a, b, g) and improve the priming of antigen-specific CD8+ T cells in vitro (Fig. 3e, f, j, k). In vivo, azido-labeled antigen-pulsed DCs also result in improved CTL response and anti-tumor efficacy compared to unlabeled DCs in both prophylactic and therapeutic settings (Fig. 4). This facile approach to improving the

**Fig. 5 | Azido-labeled DCs enable conjugation of DBCO-cytokines via click chemistry for enhanced priming of CD8+ T cells. a** Schematic illustration for the conjugation of DBCO-cytokines onto azido-labeled antigen-presenting DCs and subsequent T cell priming. **b** Percentage of Cy5-IL-2+ DCs and (**c**) mean Cy5 fluorescence intensity of DCs after incubation with DBCO/Cy5-IL-2 or Cy5-IL-2 for 30 min (*n* = 4). DCs were pretreated with Ac4ManNAz or PBS for three days. **d** Percentage of Cy5-IL-15+ DCs and (**e**) mean Cy5 fluorescence intensity of DCs after incubation with DBCO/Cy5-IL-15 or Cy5-IL-15 for 30 min (*n* = 4). DCs were pretreated with Ac4ManNAz or PBS for three days. **f** Proliferation index of OT-1 cells after 3-day coculture with IL-15 conjugated DCs (*n* = 3). DCs were pretreated with Ac4ManNAz or PBS for three days, pulsed with SIINFEKL peptide for 16 h, and incubated with DBCO-IL-15 or IL-15 (5 ng/mL) for 30 min. **g** Proliferation index of OT-1 cells after 3-day coculture with IL-2 conjugated DCs (*n* = 3). DCs were pretreated with Ac4ManNAz or PBS for three days, pulsed with SIINFEKL peptide for 16 h, and incubated with DBCO-IL-2 or IL-2 (5 ng/mL) for 30 min. **h–j** IL-15 conjugated DCs or DCs were pulsed with SIINFEKL for 16 h and then co-cultured with CFSE-stained OT-1 cells and Violet-stained non-specific CD8+ T cells for three days. **h, i** CFSE or Violet stain histogram of OT-1 cells and non-specific CD8+ T cells. **j** Proliferation index of OT-1 and non-specific CD8+ T cells (*n* = 3). Also shown are the (**k, m**) PD-1 and (**l, n**) CTLA-4 expression levels of OT-1 cells after 3-day co-culture with DCs pre-conjugated with IL-2 or IL-15 (*n* = 3). All the numerical data are presented as mean ± SD (for **b– f, g, k– m** and **n**, one-way ANOVA with post hoc Fisher's LSD test was used; for **j**, two-tailed Welch's *t*-test was used; 0.01 <*$P \le 0.05$; **$P \le 0.01$; ***$P \le 0.001$; n.s., non-significant, $P > 0.05$). Source data are provided as a Source Data file.

activation status, antigen presentation ability, and subsequent T cell priming property of DCs, via simple addition of azido-sugars to the culture medium, provides a avenue to developing enhanced DC vaccines. Compared to conventional DC-activating strategies such as the forced expression of cytokines (e.g., granulocyte-macrophage colony-stimulating factor) via viral transduction methods[19,47–49], the metabolic labeling approach is much simpler and can be easily integrated into the clinical manufacturing process of DC vaccines.

The ability to target immunomodulatory agents (e.g., cytokines) to adoptively transferred DCs in vivo holds promise to amplify CTL response and antitumor efficacy of DC vaccines, but remains challenging. Here we demonstrated that, in addition to the improved activation status, antigen presentation ability, and T cell priming property, azido-labeled DCs also enable conjugation of cytokines such as IL-2 and IL-15 via efficient click chemistry (Fig. 5b–d). While metabolic glycan labeling coupled with click chemistry has been widely used for targeted delivery of cargos to cells, as far as we know, the conjugation of cytokines onto DCs for the orchestration of DC-T cell interactions in the context of DC vaccines has not been reported before. We showed that IL-2 or IL-15 conjugated, antigen-presenting DCs result in enhanced proliferation of antigen-specific CD8+ T cells compared to the mixtures of DCs and free cytokines (Fig. 5f, g). It is noteworthy that conjugation of IL-2 or IL-15 on the surface of DCs did not induce the exhaustion of the interacting CD8+ T cells, as evidenced by the unaltered expression of inhibitory receptors such as PD-1, CTLA-3, and LAG-3 (Fig. 5k–n, Supplementary Fig. S11g, h). We further demonstrated that in vivo targeting of IL-15 to azido-labeled DC vaccines can significantly improve the CTL response and overall antitumor efficacy in both prophylactic and therapeutic settings (Fig. 6). Compared to conventional antibody targeting approaches (e.g., anti-DEC205)[50–52] that cannot distinguish adoptively transferred DCs from endogenous DCs, our approach enables specific targeting of adoptively transferred DCs. This DC labeling and targeting technology can be universally applied to all types of DC vaccines regardless of the antigen type and DC source, and can be easily adapted for different cancer types.

To conclude, we report a facile metabolic glycan labeling approach for developing enhanced DC vaccines. Azido-sugars can metabolically label DCs with azido groups, which itself can reduce the membrane mobility of DCs, activate DCs, and improve the antigen presentation and T cell priming ability of DCs. Furthermore, azido-labeled DCs enable conjugation of DBCO-cytokines including IL-2 and IL-15 via efficient click chemistry in vitro and in vivo, for improved CTL response and antitumor efficacy. This approach can be universally applied to different types of DC vaccines for treating various types of tumors. The benign safety profile also adds to the translation potential of the developed DC vaccines. The DC labeling and targeting technology not only enables the development of enhanced DC vaccines with robust CTL response and therapeutic efficacy, but also provides a platform to manipulate intercellular interactions between DCs and other immune cells.

## Methods

### Ethical statement
This research complies with all relevant ethical regulations. All procedures involving animals were done in compliance with National Institutes of Health and Institutional guidelines with approval from the Institutional Animal Care and Use Committee at the University of Illinois at Urbana-Champaign.

### Materials
D-Mannosamine hydrochloride and other chemicals were purchased from Sigma-Aldrich (St. Louis, MO, USA) unless otherwise noted. DBCO-sulfo-NHS was purchased from Click Chemistry Tools (Scottsdale, AZ, USA). DBCO-Cy5 was purchased from Thermo Fisher Scientific (Waltham, MA, USA). Fetal Bovine Serum (FBS) was purchased from Thermofisher (Waltham, MA, USA). GM-CSF, IL-2, and IL-15 were purchased from PeproTech (Cranbury, NJ, USA). SIINFEKL peptide was obtained from Peptide 2.0 (Chantilly, VA, USA). Endotoxin-free OVA was purchased from Invivogen (San Diego, CA, USA). SIINFEKL tetramer was obtained from the NIH Tetramer Core Facility (Atlanta, GA, USA). CellTrace™ CFSE Cell Proliferation Kit was purchased from Thermo Fisher Scientific (Waltham, MA, USA). Primary antibodies used in this study, including PE-conjugated anti-CD11b (Invitrogen), PE/Cy7-conjugated anti-CD11c (Invitrogen), APC-conjugated anti-CD86 (Invitrogen), PE/Cy5.5- conjugated anti-CD3-ε (Invitrogen), Alexa Fluor 700-conjugated anti-CD8-α (Invitrogen), PE/Cy7-conjugated anti-PD-1 (Invitrogen), PE-conjugated anti-CTLA-4 (Invitrogen), APC-conjugated anti-LAG-3 (Invitrogen), FITC-conjugated anti-MHCII (Invitrogen), APC-conjugated anti-CD40 (Invitrogen), and Alexa Fluor 700-conjugated anti-CCR7 (Invitrogen) were purchased from Thermo Fisher Scientific (Waltham, MA, USA). Fixable viability dye efluor780 was obtained from Thermo Fisher Scientific (Waltham, MA, USA). All antibodies were diluted according to the manufacturer's recommendations. Small molecule compounds were run on the Agilent or Shimadzu high performance liquid chromatography (HPLC). Proton and carbon nuclear magnetic resonance spectra were collected on the Varian U500 or VXR500 (500 MHz) spectrometer. Fluorescent images were taken with a EVOS microscope (Thermofisher, Waltham, MA, USA). Fluorescence measurement was conducted on a Biotek plate reader (BioTek Instruments, Winooski, VT, USA). FACS analyses were collected on Attune NxT flow cytometers and analyzed on FCS Express v6 and v7.

### Cell lines and animals
The E.G7-OVA cell line was purchased from American Type Culture Collection (ATCC CRL-2113, Manassas, VA, USA). Cells were cultured in RPMI 1640 containing 10% FBS and 100 units/mL Penicillin/Streptomycin (with 50 µg/ml G418 for E.G7-OVA cells) at 37 °C in 5% CO2 humidified air. Female C57BL/6 mice were purchased from the Jackson Laboratory (Bar Harbor, ME, USA). Feed and water were available ad libitum. Artificial light was provided in a 12 h/12 h cycle. All procedures involving animals were done in compliance with National Institutes of

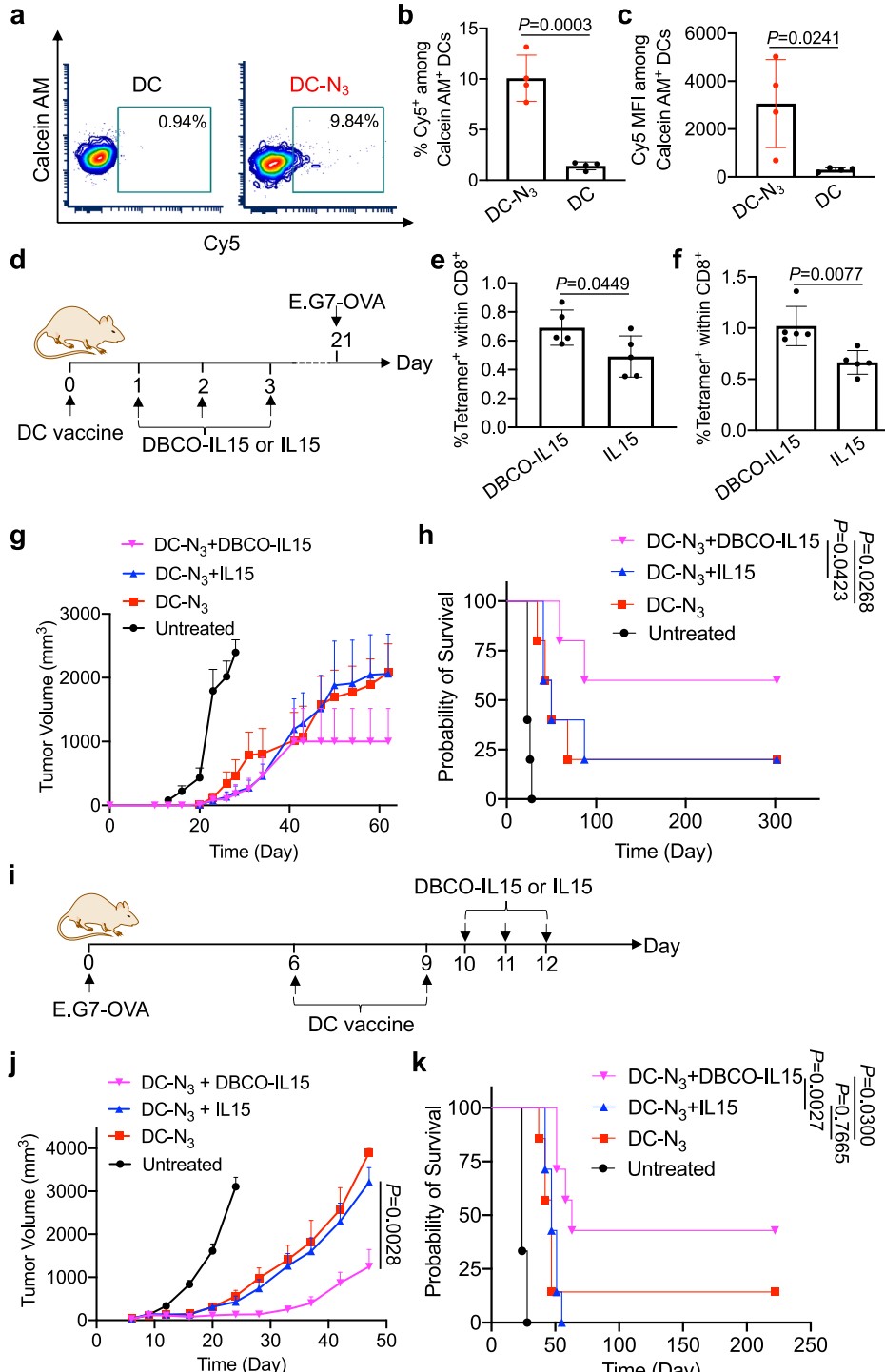

**Fig. 6 | In vivo targeting of IL-15 onto adoptively transferred DC vaccines further improves the CTL response and antitumor efficacy. a–c** DCs were pretreated with Ac₄ManNAz or PBS for three days, stained with Calcein AM, and injected into C57BL/6 mice. DBCO-Cy5 was administered 24 h later (*n* = 4). At 6 h post injection of DBCO-Cy5, lymph nodes were harvested for FACS assay. **a** Representative FACS plots of Cy5⁺ cells among Calcein AM⁺ DCs in lymph nodes. Also shown are the (**b**) percentages of Cy5⁺ cells and (**c**) mean Cy5 fluorescence intensity among Calcein AM⁺ DCs in lymph nodes. **d** Timeframe of the vaccination study. DCs were pretreated with Ac₄ManNAz for three days, pulsed with SIINFEKL peptide for 16 h, and injected into C57BL/6 mice on day 0. DBCO-IL-15 or IL-15 was injected on day 1, 2, and 3, respectively. E.G7-OVA tumor cells were

subcutaneously inoculated on day 21. Shown are the percentages of SIINFEKL-specific CD8⁺ T cells in PBMC on (**e**) day 10 and (**f**) day 20, respectively (*n* = 5). **g** Average E.G7-OVA tumor volume of each group over the course of prophylactic tumor study (*n* = 5). **h** Kaplan–Meier plots for all groups. **i** Timeframe of the therapeutic efficacy study (*n* = 5). **j** Average E.G7-OVA tumor volume of each group over the course of therapeutic tumor study (*n* = 7). **k** Kaplan–Meier plots for all groups (*n* = 7). All the numerical data are presented as mean ± SD except for **g** and **j** where data are presented as mean ± SEM (for **b**, **c**, **e**, **f**, and **j**, two-tailed Welch's *t* test was used; for **h** and **k**, one-way ANOVA with post hoc Fisher's LSD test was used, 0.01 <*P* ≤ 0.05; **P* ≤ 0.01; ****P* ≤ 0.001). Source data are provided as a Source Data file.

Health and Institutional guidelines with approval from the Institutional Animal Care and Use Committee at the University of Illinois at Urbana-Champaign.

## Synthesis of Ac₄ManNAz

D-Mannosamine hydrochloride (1.0 mmol) and triethylamine (1.0 mmol) were dissolved in methanol, followed by the addition of $N$-(2-azidoacetyl) succinimide (1.2 mmol). The mixture was stirred at room temperature for 24 h to yield $N$-(2-azidoacetyl)mannosamine, which was directly used for the next step without purification. Solvent was removed under reduced pressure and the residue was re-dissolved in pyridine. Acetic anhydride was added and the reaction mixture was stirred at room temperature for another 24 h. After removal of the solvent, the crude product was purified by silica gel column chromatography using ethyl acetate/hexane (1/1, v/v) as the eluent to yield a white solid (1/1 α/β isomers). $^1$H NMR (CDCl₃, 500 MHz): δ (ppm) 6.66&6.60 (d, $J$ = 9.0 Hz, 1H, C(O)N$H$CH), 6.04&6.04 (d, 1H, $J$ = 1.9 Hz, NHCHC$H$O), 5.32-5.35&5.04-5.07 (dd, $J$ = 10.2, 4.2 Hz, 1H, CH₂CHC$H$CH), 5.22&5.16 (t, $J$ = 9.9 Hz, 1H, CH₂CHCHC$H$), 4.60-4.63&4.71-4.74 (m, 1H, NHC$H$CHO), 4.10-4.27 (m, 2H, C$H_2$CHCHCH), 4.07 (m, 2H, C(O)C$H_2$N₃), 3.80-4.04 (m, 1H, CH₂C$H$CHCH), 2.00-2.18 (s, 12H, C$H_3$C(O)). $^{13}$C NMR (CDCl₃, 500 MHz): δ (ppm) 170.7, 170.4, 170.3, 169.8, 168.6, 168.3, 167.5, 166.9, 91.5, 90.5, 73.6, 71.7, 70.5, 69.1, 65.3, 65.1, 62.0, 61.9, 52.8, 52.6, 49.9, 49.5, 21.1, 21.0, 21.0, 20.9, 20.9, 20.9, 20.8. ESI MS ($m/z$): calculated for C₁₆H₂₂N₄O₁₀Na [M+Na]$^+$ 453.1, found 453.1.

## General procedures for flow cytometry analysis of azido-labeled dendritic cells

Bone marrow cells were extracted from the tibia and femur of C57BL/6 mice and cultured in RPMI medium containing GM-CSF. On day 6, cells were seeded in a 24-well plate at a density of $4 \times 10^4$ cells per well. Ac₄ManNAz was added and incubated with cells for 72 h. After washing, cells were stained with DBCO-Cy5, anti-CD11b, and anti-CD11c at 4 °C for 20 min, prior to FACS analysis. For DC activation analysis, cells were stained with fluorophore-conjugated anti-CD86 and anti-MHCII prior to FACS analysis.

## Confocal imaging of DCs

Day-6 BMDCs were seeded in a 4-chamber slide at a density of $4 \times 10^4$ cells per well. Ac₄ManNAz was added and incubated with cells for 72 h. After washing, cells were incubated with DBCO-Cy5 for 20 min. Cells were then fixed in 4% PFA, stained with DAPI for 10 min, and imaged under a confocal microscope.

## Fluorescence recovery after photobleaching (FRAP) assay

DCs were transfected with the ICAM-1-GFP plasmid or YFP-Mem plasmid to express GFP-tagged ICAM-1 or YFP-Mem on the cell membrane. After transfection, DCs were incubated with Ac₄ManNAz or PBS for 72 h. After washing for four times, cells in a culture dish were placed under a confocal microscope in a live cell imaging chamber. Before the bleaching step, regions of interest (ROIs) on cells were drawn and pre-bleaching images of cells were taken. The selected region was then bleached by turning on the laser at 100% transmission, followed by a time-series image capture. Fluorescence intensity (after normalization from prebleach images) of ROIs was determined and plotted against time. Mobile fraction and half time of recovery were then calculated from the fitted curves.

## Stability of cell-surface azido groups

BMDCs were pretreated with Ac₄ManNAz for three days and then transferred to fresh media. At 3, 6, 12, 24, 48, 72, and 96 h, respectively, cells were incubated with DBCO-Cy5 for 30 min and stained with fluorophore-conjugated anti-CD11c and live/dead stain to analyze cell-surface azido groups.

## Membrane retention of conjugated molecules

BMDCs were pretreated with Ac₄ManNAz for three days, and incubated with DBCO-Biotin for 30 min. Cells were then transferred to fresh media. At 3, 6, 12, 24, 48, 72, and 96 h, respectively, cells were incubated with FITC-avidin and fluorophore-conjugated anti-CD11c prior to flow cytometry analysis.

## Antigen presentation by DCs

BMDCs were pretreated with Ac₄ManNAz or PBS for three days, and then incubated with the SIINFEKL peptide or OVA protein of varying concentrations for 16 h in the presence of Ac₄ManNAz. The expressed SIINFEKL on the surface of DCs was detected by APC-conjugated anti-MHCI-SIINFEKL via flow cytometry. Cells were also stained with fluorophore-conjugated anti-CD86 and anti-MHCII for the analysis of DC activation status.

## Coculture of DC and OT-1 cells

To further validate the presentation of SIINFEKL antigen by DCs, DCs treated with OVA protein or SIINFEKL peptide were incubated with CFSE-stained OT-1 cells (1/1 T cell/DC ratio) for three days. FACS assay was performed to analyze the proliferation index of OT-1 cells.

## In vitro conjugation of cytokines onto azido-labeled DCs

DBCO-IL-15 was synthesized via the conjugation of DBCO-sulfo-NHS with IL-15, and purified via ultracentrifugation with an Amicon filter (3 kDa cut-off molecular weight). DBCO/Cy5-IL-15 was synthesized via conjugation of Cy5-NHS and DBCO-sulfo-NHS to IL-15. Similarly, DBCO-IL-2 was synthesized via the conjugation of DBCO-sulfo-NHS with IL-2, and purified via ultracentrifugation. DBCO/Cy5-IL-was synthesized via conjugation of Cy5-NHS and DBCO-sulfo-NHS to IL-2. For surface display of cytokines, BMDCs were pretreated with azido-sugars or PBS for three days and then incubated with DBCO/Cy5-cytokines or Cy5-cytokines for 30 min, prior to FACS assay.

## Co-culture of cytokine-conjugated DCs and OT-1 cells

BMDCs were treated with Ac₄ManNAz for 72 h, pulsed with SIINFKL peptide for 16 h, and incubated with DBCO-IL-15 or IL-15 or DBCO-IL-2 or IL-2 for 30 min. Cytokine-conjugated DCs were then cocultured with CFSE-stained OT-1 cells (1/1 T cell/DC ratio) for three days. FACS assay was performed to analyze the proliferation index of OT-1 cells. For some experiments, violet stain-stained non-specific CD8$^+$ T cells were incubated together with OT1 cells and BMDCs (1/1/1 OT1 cell/non-specific T cell/DC ratio) for three days. FACS assay was performed to analyze the proliferation index of OT-1 cells and non-specific CD8$^+$ T cells.

## In vivo DC targeting study

Female C57BL/6 mice (5-7 weeks) were subcutaneously injected (lower flank) with Calcein AM-stained DCs (with or without 3-day Ac₄ManNAz pretreatment), followed by subcutaneous injection (distant from DC injection site) of DBCO-Cy5 after 24 h. At 6 h post injection of DBCO-Cy5, lymph nodes were harvested. Cells from lymph nodes were collected and stained with anti-CD11b, anti-CD11c, and live/dead stain, prior to FACS analysis.

## Vaccination study with azido-labeled DC vaccines

Female C57BL/6 mice (5-7 weeks) were divided into 3 groups ($n$ = 5-6 per group): azido-labeled DC vaccine; control DC vaccine; and untreated. DC vaccines were prepared by pretreating BMDCs with Ac₄ManNAz or PBS for three days and pulsing with SIINFEKL peptide for 16 h. DC vaccines ($1 \times 10^6$ per mouse) were subcutaneously injected into the flank of mice on day 0. On day 10 or 20, blood was collected for the analysis of SIINFEKL-specific CD8$^+$ T cells in PBMCs: red blood cells were lysed and cell pellets were stained with APC-conjugated H2Kb-SIINFEKL tetramer, prior to FACS analysis.

For prophylactic tumor study, mice were challenged with a subcutaneous injection of $2.5 \times 10^5$ E.G7-OVA cells on day 21. Tumor growth and body weight of animals were closely monitored. The tumor volume was calculated using the formula (length) × (width)$^2$/2, where the long axis diameter was regarded as the length and the short axis diameter was regarded as the width. Mice were euthanized when the largest diameter of tumors reaches 20 mm or mice became moribund.

## Therapeutic tumor study with azido-labeled DC vaccines

Female C57BL/6 mice (5–7 weeks) were divided into three groups ($n = 6$-7 per group): azido-labeled DC vaccine; control DC vaccine; and untreated. E.G7-OVA cells ($5 \times 10^5$ cells per mouse) were subcutaneously injected into the upper right flank of C57BL/6 mice on day 0. On day 7, DC vaccines were subcutaneously injected into the lower right flank of mice. In some studies, a second dose of DC vaccines was given on day 14. Tumor growth and body weight of animals were closely monitored. The tumor volume was calculated using the formula (length) × (width)$^2$/2, where the long axis diameter was regarded as the length and the short axis diameter regarded as the width. Mice were euthanized when the largest diameter of tumors reaches 20 mm or mice became moribund.

## Vaccination study involving IL-15 targeting

Female C57BL/6 mice (5–7 weeks) were divided into 4 groups ($n = 5$-6 per group): azido-labeled DC vaccine + DBCO-IL-15; azido-labeled DC vaccine + IL-15; azido-labeled DC vaccine; and untreated. DC vaccines were prepared by pretreating BMDCs with $Ac_4ManNAz$ or PBS for three days and pulsing with SIINFEKL peptide for 16 h. DC vaccines ($1 \times 10^6$ cells per mouse) were subcutaneously injected into the lower flank of mice on day 0. DBCO-IL-15 or IL-15 were subcutaneously injected (with a distance from DC injection site) once daily for three days (Day 1, 2, and 3). At selected times, blood was collected for the analysis of SIINFEKL-specific CD8$^+$ T cells in PBMCs via tetramer staining and FACS assay. For prophylactic tumor study, mice were challenged with a subcutaneous injection of $2.5 \times 10^5$ E.G7-OVA cells. Tumor growth and body weight of animals were closely monitored. The tumor volume was calculated using the formula (length) × (width)$^2$/2, where the long axis diameter was regarded as the length and the short axis diameter was regarded as the width. Mice were euthanized when the largest diameter of tumors reaches 20 mm or mice became moribund.

## Therapeutic tumor study involving IL-15 targeting

Female C57BL/6 mice (5-7 weeks) were divided into three groups ($n = 7$ per group): azido-labeled DC vaccine + DBCO-IL-15; azido-labeled DC vaccine + IL-15; azido-labeled DC vaccine; and untreated. E.G7-OVA cells ($5 \times 10^5$ cells per mouse) in HBSS were subcutaneously injected into the upper right flank of C57BL/6 mice on day 0. On day 6, DC vaccines were subcutaneously injected into the lower right flank of mice, followed by a second dose of DC vaccines on day 9. DBCO-IL-15 or IL-15 was subcutaneously injected once daily on day 10, 11, and 12, respectively. Tumor growth and body weight of animals were closely monitored. The tumor volume was calculated using the formula (length) × (width)$^2$/2, where the long axis diameter was regarded as the length and the short axis diameter was regarded as the width. Mice were euthanized when the largest diameter of tumors reaches 20 mm or mice became moribund.

## Construction of strand-specific RNAseq libraries

Extraction of total RNA, construction of libraries and sequencing on the Illumina NovaSeq 6000 were performed at the Roy J. Carver Biotechnology Center at the University of Illinois at Urbana-Champaign. Total RNA was extracted from dendritic cells was done with the RNeasy Plus kit from Qiagen. Total RNAs were run on a Fragment Analyzer (Agilent) to evaluate RNA integrity. RNAseq libraries were constructed with the Kapa Hyper Stranded mRNA Sample Prep kit (Roche). Briefly, polyadenylated messenger RNAs (mRNAs) were enriched from 500 ng of high quality DNA-free total RNA with oligodT beads. The mRNAs were chemically fragmented, annealed with a random hexamer and converted to double stranded cDNAs, which were subsequently blunt-ended, 3'-end A-tailed and ligated to indexed adaptors. Each library was ligated to a uniquely dual indexed adaptor (unique dual indexes) to prevent index switching. The adaptor-ligated double-stranded cDNAs were amplified by PCR for 8 cycles with the Kapa HiFi polymerase (Roche) to reduce the likeliness of multiple identical reads due to preferential amplification. The final libraries were quantitated with Qubit (ThermoFisher) and the average library fragment length was determined on a Fragment Analyzer. The libraries were diluted to 10 nM and further quantitated by qPCR on a CFX Connect Real-Time qPCR system (Biorad) for accurate pooling of the barcoded libraries and maximization of number of clusters in the flowcell.

## Sequencing of libraries in the NovaSeq

The pooled barcoded RNAseq libraries were loaded on a NovaSeq SP lane for cluster formation and sequencing. The libraries were sequenced from one end of the fragments for a total of 100nt from. The fastq read files were generated and demultiplexed with the bcl2fastq v2.20 Conversion Software (Illumina).

## RNA-seq data analysis

Sequenced reads were pseudo-aligned to NCBI's *Mus musculus* Annotation Release 109 transcripts using Salmon (v1.10.0) and the GRCm39 genome as the decoy sequence and with the following options: --seqBias --gcBias --numBootstraps = 30 --validateMappings --recoverOrphans. The transcript-levels counts were then read into R (v.4.3.0) and gene-level counts were calculated using the "lengthScaledTPM" method from the tximport package (v1.28.0); this method provides more accurate gene-level counts estimates and keeps multi-mapped reads in the analysis compared to traditional alignment-based method. Genes that did not have at least 0.25 counts per million mapped reads (CPM) in at least 4 samples were filtered out. Additional normalization to correct for RNA composition was done using the TMM method in the edgeR package (v3.42.2). edgeR's quasi-likelihood method was used to find genes that had fold-changes significantly larger than +/−2 fold-change. Multiple test correction was done using the False Discovery Rate method.

## Statistical analyses

Statistical analysis was performed using GraphPad Prism v6 and v8. Sample variance was tested using the F test. For samples with equal variance, the significance between the groups was analyzed by a two-tailed Student's $t$ test. For samples with unequal variance, a two-tailed Welch's t-test was performed. For multiple comparisons, a one-way analysis of variance (ANOVA) with post hoc Fisher's LSD test was used. The results were deemed significant at $0.01 <*P \leq 0.05$, highly significant at $0.001 <**P \leq 0.01$, and extremely significant at $***P \leq 0.001$.

## Statistics & reproducibility

The sample sizes were determined empirically. In general, in vitro studies involve $n = 3$–6 and in vivo studies involve $n = 5$–8. No data were excluded from the analyses. Randomization was used for animal studies.

## Reporting summary

Further information on research design is available in the Nature Portfolio Reporting Summary linked to this article.

## Data availability

All data provided in this study can be found in the main text, figures, supplementary information, and source data files. RNA sequencing data generated in this study are available in the GEO database under accession code GSE238016. Source data are provided with this paper.

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

## Acknowledgements

The authors would like to acknowledge the financial support from NSF DMR 21-43673 CAR (H.W.), NIH R01CA274738 (H.W.), NIH R21CA270872 (H.W), and start-up package (H.W.) from the Department of Materials Science and Engineering at the University of Illinois at Urbana-Champaign and the Cancer Center at Illinois (CCIL). J.H. acknowledges the support from the Cancer Scholars for Translational and Applied Research (C*STAR) Program sponsored by the Cancer Center at Illinois and the Carle Cancer Center under Award Number CST EP012023. R.B. acknowledges the support from the National Institute of Biomedical Imaging and Bioengineering of the National Institutes of Health under Award Number T32EB019944. We would also like to acknowledge the help from Alvaro Hernandez and Jenny Drnevich at UIUC Roy J. Carver Biotechnology Center on mRNA sequencing and transcriptome analy-sis. A.E.A. acknowledges the support from the European Research Council (ERC) under the European Union's Horizon 2020 research and innovation programme (grant agreement number 851055). A.E.A. is supported by the Francis Crick Institute, which receives its core funding from Cancer Research UK (CC2214), the UK Medical Research Council (CC2214), and the Wellcome Trust (CC2214).

## Author contributions

J.H. and H.W. conceived the study, designed the experiments and wrote the manuscript. J.H., R.B., Y.L., Y.B., and A.E.A. carried out the experi-ments and contributed to the revision of the manuscript.

## Competing interests

The authors declare no competing interests.
