## [Peer Review File · Nature Communications]

REVIEWER COMMENTS

Reviewer #1 (Remarks to the Author):

The manuscript “Metabolic Glycan Labeling Immobilizes Dendritic Cell Membrane and Enhances Antitumor Efficacy of Dendritic Cell Vaccine” (NCOMMS-22-51490A-Z) by Hua Wang and coauthors is a well written, carefully edited manuscript that describes measurable increases in DC vaccine efficacy in vivo. The manuscript has several significant strong points (as described below) but also has a few holes, mostly at the molecular / mechanistic level (also as described below). In my opinion these holes will need to be filled in to gain a mechanistic understanding of the system to move it forward into clinical translation but I will not take a stand on whether these current shortcomings need to be addressed experimentally for this particular study to be published in Nature Communications or not (I will leave that to the editor). There are a number of fairly minor points that need to be addressed before publication either way (also as listed below)

Noteworthy results: This manuscript describes the metabolic glycan labeling of dendritic cells (DCs) with Ac4ManAz, an azido-labeled precursor for sialic acid biosynthesis. The glycan labeling method alone had a beneficial impact on DC activation and also provided a chemical tag to conjugate immunomodulatory moieties onto the surfaces of DCs (e.g., IL-2 and IL-15) that further enhanced DC activity both in vitro and in an in vivo tumor model. The mechanism proposed to explain the impact of “stand alone” sugar analog treatment was that the membrane mobility was reduced, prolonging the surface residence times of activation markers.

Significance of results: DC cancer vaccine development has lagged making the “stand alone” use of Ac4ManAz an attractive, potentially easily translatable way to improve this class of cancer therapeutics. (The further “bells and whistles” [i.e., conjugation of cytokines to metabolically glycoengineered cells is likely to be less easily translated but the improvement in DC vaccine efficacy makes efforts to do so a worthy goal]).

Support for conclusions and claims: Overall the conclusions and claims that are made in this manuscript are supported by the data shown. One possible exception is the claims that emanate from the FRAP analysis, which is rather elegant btw. This assay shows quite convincingly that membrane fluidity is decreased by azido-analog treatment, which in turn increases the surface

residence time of markers associated with DC activation; this is shown indirectly with ICAM-1GFP and YFP-Mem. These results were extrapolated to CD86 levels (e.g., to explain higher levels of this marker on azido analog treated cells); a fairly easy to do experiment to further support or debunk this idea is transcript analysis of CD86 (and other relevant markers). Has this been done to show whether the analog affects transcription or transcript stability?

Soundness of methodology:

A weak point of the study is a lack of explicit acknowledgment that Ac4ManAz modifies cell surface sialic acids (i.e., it installs azido-modified sialic acids on the cell surface). Therefore the claims that azido presentation alone on the cell surface has an impact on DC biology should be presented in this context with appropriate controls. For example, the requirement for the azido group to be presented as a sialoside could/should be tested by evaluating other analogs used for metabolic glycan labeling such as azido-modified GalNAc or azido-modified fucose; these experiments would help unravel whether sialic acid per se is needed.

Another way to address the role of sialic acid is to consider that sialylation is a regulator of galectin lattice interactions, a topic that has been studied by FRAP in the context of EGFR trafficking (e.g., doi: 10.1083/jcb.200611106 and doi: 10.18632/oncotarget.11582); note that the latter study modulates sialylation through ManNAc analog treatment conceptually similar to the experiments covered in this study. In this study an appropriate control analog would be Ac4ManNAc, which increases flux into the sialic acid pathways to test if increased sialylation has an effect on the DC endpoints measured in this study (presumably Ac4ManNAz both increases flux through the sialic acid pathway as well as introducing non-natural sialic acids onto the cell surface; this study does not address which of these two effects is responsible [or more highly responsible, they likely both have a role]). To follow up, if Ac4ManNAc, which increases cell surface display of Neu5Ac (the predominant form of this sugar in humans), does not elicit the DC endpoints covered in this study, do other non-natural ManNAc analogs [I'm guessing that there are at least 20 of them reported] have this effect or is it unique to azido analogs?)

Useful information to include is the "washout" rate of cell surface azido groups; i.e., what is the decay rate after removal of Ac4ManAz? I suspect that it is substantial during the 16 hour incubations when tagging the cells with peptide or ILs. Note that this comment is not intended to

be critical of the loss of cell surface labeling per se; rather the authors could make the point that the labeling, although transient of itself, sets a durable set of downstream responses in motion of (potential) therapeutic value).

Another methodology-related issue is that no explanation for why P400 was used is given. For example, was it shown to be a better labeling reagent than Ac4ManAz in previous studies (in the present study it didn't seem any better and at some point seemed to be dropped from the experiments with no clear explanation why that was done either)? Also, the way the concentrations are presented comparing Ac4ManAz and P400 as given don't seem plausible (i.e., that both were used at 200 μ M or 400 μ M; does this mean that ~400-fold the molar equivalent of ManAz was delivered in the P400 experiments considering that each copy of P400 supplies 400 ManAz's? (I assume the dose was adjusted according to supply the same equivalents but this doesn't seem to be explained anywhere, it should be, and the figures need to be labeled to address this issue as well).

Detail of methods:

The FRAP assay needs more detail; in particular, how many washing steps were done, e.g., to remove residual FBS and what were the exact media conditions used? This is important because any residual FBS could supply an exogenous source of galectins, which could affect the results (the role of galectin and sialic acids has been raised above)

This point was also alluded to above, but the methods do not explicitly describe the experiments where the DCs were treated with analog for 72 h and then treated subsequently for 16 h (or other time points) with the SIINFEKL peptide (or whatever). Was Ac4ManAz maintained in the system during these 16 hours? (I'm assuming not because although its inclusion would help maintain cell surface levels azido moieties, the free analog would compete for conjugate formation, preventing facile surface labeling [just curious, have experiments actually been done to see which effect would be greater?])

Minor points:

Throughout: is the term "membrane mobility" shorthand for "mobility of proteins in (the) membrane"? To me the authors are more accurately / precisely referring to "membrane fluidity" (i.e., the viscosity of the lipid bilayer and how this affects the movement of membrane components)

Page 2, last sentence of first paragraph: should "the benign safety profile" be "their benign safety

profile”?

Page 2, second paragraph: should “explored as DC source” be “explored as a DC source” or “explored as the DC source”?

Page 2, second paragraph: does the word “could” (in “could undermine the resultant CTL”) imply that the rest of the sentence is just hypothetical? (for example, compare the meaning of the sentence if “often” or “are factors that” were used instead of could).

Page 3, RESULTS section (and throughout): “Ac4ManAz” is more commonly (and accurately) written as “Ac4ManNAz” (i.e., adding the “N” before Az makes it explicit where the azido group occurs in the molecule).

Page 5: The sentence that includes “SIINFEKL, considering the well-established characterization methods” should include a reference to these methods

Page 5, a few lines later: Would just “Consistent with” be better than “In consistence with”?

Page 12, “Cell lines and animals” section: Penicillin/Streptomycin do not need to be capitalized and CO2 should be CO₂. Where was the FBS purchased from (perhaps this is listed in the Materials section but I’m not finding it)?

Page 12, Synthesis of Ac4Man(N)Az: The synthesis of this compound is “well known” and probably doesn’t need to be described in detail (if the journal has no space limitations, I suppose that inclusion of this information is fine) unless the authors use a revised/new synthetic route (did they?)

Page 13, second paragraph: A more descriptive name for “Ac3Man(N)AzOH” would be 3,4,6-OAc3ManNAz (i.e., the locations of the acetyl groups would be specified).

Page 16, first line (and near the end of the page): To be clear on the dose of “DC vaccines (1X10⁶ per mouse)”, does that mean 1X10⁶ cells per mouse?

Page 21, Figure 1 legend: (note that this point also arises in the main text). What does “targeted conjugation” mean? As the authors point out here in the figure legend, azido-sugars occur on glycoproteins and glycolipids, which is indeed correct, and furthermore, occur on many/most cell surface glycoproteins and glycolipids. From this perspective, the cell surface labeling of membrane components is rather ubiquitous and not really targeted per se. To me, replacing “targeted conjugation” with something like “cell surface conjugation” would be considerably more accurate.

Also, considering that the authors mention the glycolipids can be modified with the azido-bearing sugars, can (should?) this be shown in their cartoons? (considering that lipid / membrane fluidity is affected, this could be directly relevant...)

Page 23, Figure 2 legend: "ate" should be "are" (typo) in (d); in (g) "after treated" should be "after treatment" or "after being treated"

Page 25, Figure 3 legend: The term "Ac4" shown on the figure should be defined in the figure legend.

Reviewer #2 (Remarks to the Author):

Dear Authors,

Han et al., report the metabolic glycan labelling on dendritic cells improved the antigen presentation and antitumor immune response and demonstrate the application of the approach to target conjugate useful cytokines to improve the effector cell response. Earlier studies demonstrated the potential of a gel based in situ labelling approach to target the DCs and take the advantage of click chemistry to label cytokines of interest to improve the T cell expansion. Current study focus on the in vitro labelling and vaccination of antigen loaded dendritic cells as vaccine for cancer immunotherapy. Authors demonstrated the efficacy of the approach using in vivo models and 50% of the vaccinated animals shows a long-term survival. In general, the study is well conducted and there are some concerns as listed below.

The study shows an effective antitumor immune response and the underlying mechanism is not well defined. Authors claims the treatment with Ac4ManAz or P400 improved DC activation (Fig 1) based on the expression of CD86 and MHC II. It will be great to assess the expression of additional markers such as CD40, CD83 or CCR7 to confirm there is a really a DC activation. The cytokine profile does not show any significant difference on IL6 or IL-12 to confirm the treatments activate the DCs (Fig S3). The observed change in CD86 and MHC may be due to the glycosylation of the molecules due to the glycan treatment and that may reduce the ubiquitination and turnover rate of these molecules. The same concept is applicable for the ICAM-1 and the observed stability may be due to the reduced turnover rate and lack of ubiquitination of the ICAM-1 molecule.

Authors claims that Ac4ManAz treated DCs have advantage on antigen processing and presentation shows a better T cell activation and antigen presentation. The improved T cell activation can be directly correlated to the high MHC molecules on treated DCs, than antigen processing. If treated and untreated molecules have a similar level of MHC and there is difference in antigen presentation there can be an inherent difference in their antigen processing ability. It will be great to understand what is the advantage Ac4 treated DCs compared to untreated DCs to elicit the antitumor immune response? Also the contribution of tissue resident DCs as well as other immune cells such as NK cells in the observed tumor regression. Nanoparticle or VLPs decorated with saccharides demonstrated an improved lymph node draining and improved the antigen specific T cell response. The AC4 treated DCs may have an improved lymph node draining and may be the tissue resident DCs may be capturing the antigens

through cross-dressing and activate T cells. It is critical to evaluate the direct and indirect antigen presentation in this context to understand the how exactly the AC4 treated DCs improve the T cell response. A recent study clearly demonstrated that the BMDCs are incapable of direct antigen presentation and testing the vaccination in a BATF3^{-/-} or IRF8^{32^{-/-}} mouse model will be critical to understand the contribution of lymph node resident DCs in T cell activation. The AC4 treated DCs may have an advantage to carry the antigen to the draining lymph node due to glycan treatment and results in the better presentation. The authors should address the underlying mechanism and it is important for translational settings where the endogenous cDC1s in cancer patients are generally compromised in their frequency and function.

In vivo targeting of IL-15 is an interesting approach and it will be great to show dot plots with other cells including in situ DCs to demonstrate that the DBCO binding is specific for injected DCs. It is also interesting to check the frequency of NK cells upon the IL-15 injection.

Other

Prophylactic tumor study (Figure 4) : the Schema shows the tumor cells are implanted on day 21 and Figure 4d. shows a tumor growth on day 10.

Reviewer #3 (Remarks to the Author):

In this manuscript, Han et al reported a metabolic labeling approach for the targeted delivery of T-cell-stimulating cytokines to the adoptively transferred DC vaccine. Interestingly, they show that metabolic glycan labelling itself enhances DC maturation, antigen cross-presentation. Combination of labeled DC vaccine and in vivo administered cytokines improve the anticancer efficacy. This chemical-conjugation-based approach is interesting and provides a new way to target transferred DCs in vivo. The in vitro and in vivo experiments were well designed to demonstrate the targeting effects. Most interpretations were reasonable and conclusions were supported by experimental evidence. There are several issues that should be addressed to improve the current manuscript.

1. it is interesting that the authors found that metabolic glycan labelling itself can enhance maturation of DCs. However, this phenomenon needs to be further confirmed with additional experiments: 1) controls groups including sugar alone (no azide group), polymer alone (no sugar side chains), and sugar-azide monomer should be added to investigate which part indeed contributes to the observed effect; 2) Fig 2g-i, it is possible that the fluorescence of Cy5 bleeds to other channels. To exclude that possibility, the

author may use the same color (as the one for CD86) to label a non-changed marker (e.g., CD45) and show no difference between Cy5+ and Cy5- groups; 3) other maturation markers including CD80, MHC-II should be assessed (ideally with different colors, for the reason of 2)

2. The mechanism underlying the enhanced maturation/antigen presentation of DC by metabolic glycan labelling was not clear. The authors showed the results of photobleaching for a measurement of membrane fluidity. However, the impact on membrane fluidity seems minimum (Fig 2j, k; Figure S4), and whether it is a causation (rather than a correlation) remains to be determined. This mechanistic study is weak and better removed.

3. Fig 2: it is not clear why the authors investigated P400 here. If I understand correctly, for all the rest of the results (Fig 3-6), only Ac4ManAz was used for metabolic labeling of DCs. If so, the data of P400 are not relevant.

4. Fig 3: the data for SIIFEKL or OVA protein are similar, please keep one and move the other to SI. Please use consistent colors to label groups w or w/o Ac4. Why different concentrations of OVA were used in j and k?

5. From the OT-1 CD8 T cell proliferation data shown in Fig 3, it seems that metabolic glycan labelling enhances antigen cross-presentation. What is the possible mechanism, endosomal escape of the antigens? Please provide evidence for possible mechanisms.

6. Fig 5f, g: what if same amount of IL-15 (or IL-2) was added directly to the co-culture rather than pretreatment of DCs with cytokines?

7. The chemical approach for targeting cytokines to transferred DCs is certainly interesting and new. However, there are several issues: 1) how long can the surface-bound cytokine stay? DCs also express the common cytokine receptor gamma chain including IL-2 receptor (IL-2R) (<https://academic.oup.com/intimm/article/10/2/237/674045>). Once bind to the cytokine receptor, the surface cytokines may be quickly internalized by DCs and therefore have no effects on the T cells engaged; 2) the cytokines presented on DC surface still have the potential to trigger activation of tumor-antigen non-specific T cells. How much gain in safety as compared to administration of equivalent cytokine remains to be determined; 3) when DCs are properly activated, they naturally secrete cytokine as signal 3 to boost T cells engaged. Additional surface-bound cytokine may have minimum further improvement in this action.

8. Fig 6d-f: a better experiment to do is to transfer of DC vaccine together with naive OT-1 T cells in vivo, and compare expansion of OT-1 vs. endogenous T cells (as antigen non-specific) in mice receiving cytokines or DBCO-cytokines. DBCO-cytokines are expected to boost more OT-1 T cells but less endogenous T cells.

9. Fig 6g,h: tumor growth was almost identical but why the survival curves differ so much?

Minor issues:

1. For the same flow cytometry data, showing either frequency, counts, or MFI is enough. No need to duplicate the same data with two different forms of data presentation (Fig 1, 2)
2. The endpoints (tumor size, health condition, etc) for mouse tumor models studies should be stated in the method part.

Point-by-point reply to reviewers comments

(All responses were colored in blue, and all changes in the manuscript were highlighted in yellow)

Reviewer #1

The manuscript “Metabolic Glycan Labeling Immobilizes Dendritic Cell Membrane and Enhances Antitumor Efficacy of Dendritic Cell Vaccine” by Hua Wang and coauthors is a well written, carefully edited manuscript that describes measurable increases in DC vaccine efficacy in vivo. The manuscript has several significant strong points (as described below) but also has a few holes, mostly at the molecular / mechanistic level (also as described below). In my opinion these holes will need to be filled in to gain a mechanistic understanding of the system to move it forward into clinical translation but I will not take a stand on whether these current shortcomings need to be addressed experimentally for this particular study to be published in Nature Communications or not (I will leave that to the editor). There are a number of fairly minor points that need to be addressed before publication either way (also as listed below)

Response: We thank the reviewer for the positive comments, and have addressed the questions accordingly below.

Noteworthy results: This manuscript describes the metabolic glycan labeling of dendritic cells (DCs) with Ac4ManAz, an azido-labeled precursor for sialic acid biosynthesis. The glycan labeling method alone had a beneficial impact on DC activation and also provided a chemical tag to conjugate immunomodulatory moieties onto the surfaces of DCs (e.g., IL-2 and IL-15) that further enhanced DC activity both in vitro and in an in vivo tumor model. The mechanism proposed to explain the impact of “stand alone” sugar analog treatment was that the membrane mobility was reduced, prolonging the surface residence times of activation markers.

Response: We truly appreciate the great summary from the Reviewer.

Significance of results: DC cancer vaccine development has lagged making the “stand alone” use of Ac4ManAz an attractive, potentially easily translatable way to improve this class of cancer therapeutics. (The further “bells and whistles” [i.e., conjugation of cytokines to metabolically glycoengineered cells is likely to be less easily translated but the improvement in DC vaccine efficacy makes efforts to do so a worthy goal]).

Response: Thanks for highlighting the significance of our results.

(1) Support for conclusions and claims: Overall the conclusions and claims that are made in this manuscript are supported by the data shown. One possible exception is the claims that emanate from the FRAP analysis, which is rather elegant btw. This assay shows quite convincingly that membrane fluidity is decreased by azido-analog treatment, which in turn increases the surface residence time of markers associated with DC activation; this is shown indirectly with ICAM-1GFP and YFP-Mem. These results were extrapolated to CD86 levels (e.g., to explain higher levels of this marker on azido analog treated cells); a fairly easy to do experiment to further support or

debunk this idea is transcript analysis of CD86 (and other relevant markers). Has this been done to show whether the analog affects transcription or transcript stability?

Response: We thank the Reviewer for the positive comments. We have now supplemented the transcriptome analysis by performing mRNA sequencing of DCs treated with Ac₄ManNAz or PBS. Strikingly, we observed the radically changed gene expression in Ac₄ManNAz-treated DCs versus untreated DCs, with 1947 genes upregulated and 2949 genes downregulated (**Fig. 2j, Fig. S5**). The significantly upregulated genes include H2-K2, H2-D1, IL-15, IL-1a, TNF-a (**Fig. S6**), which is consistent with the improved antigen presentation and inflammatory phenotype of DCs. While CD86 did not show a significant change in mRNA levels, we did observe the upregulated expression of spic (spi1/PU.1 related), a transcription factor regulating the translation of CD86 (**Fig. S6f-g**).

We have now added **Fig. 2j, Fig. S5, Fig. S6**, and the following text “To better comprehend the potential mechanism behind the upregulation of activation markers observed with metabolic glycan labeling, we also performed the transcriptome analysis of Ac₄ManNAz- and PBS-treated DCs. Strikingly, Ac₄ManNAz treatment resulted in radical and widespread transcriptional changes of DCs, with 1,947 genes upregulated and 2,949 genes downregulated (**Fig. 2j, Fig. S5, Fig. S6**). The significantly upregulated genes include H2-K2, H2-D1, IL-15, IL-1a, TNF-a (**Fig. S6a-e**), which is consistent with the improved antigen presentation and inflammatory phenotype of DCs. While CD86 did not show a significant change in mRNA levels, we did observe the upregulated expression of spic (spi1/PU.1 related), a transcription factor regulating the translation of CD86 (**Fig. S6f-g**)” in the revised manuscript.

Fig. 2j. Altered gene expression profiles between Ac₄ManNAz-treated DCs and untreated DCs. DCs were treated with 200 μ M Ac₄ManNAz or PBS for three days.

Fig. S5. Heatmap of 20 most significant up and down regulated genes from DCs treated with Ac₄ManNAz (200 μM) or PBS for three days.

Fig. S6. Metabolic glycan labeling radically alters the gene expression of DCs. Normalized expression of (a) H2-K2 (MHCI), (b) H2-D1 (MHCI), (c) IL-15, (d) IL1α, (e) TNF-α, (f) CD86, and (g) Spi-C mRNAs, as determined by the transcriptome analysis.

(2) Soundness of methodology: A weak point of the study is a lack of explicit acknowledgment that Ac₄ManAz modifies cell surface sialic acids (i.e., it installs azido-modified sialic acids on the cell surface). Therefore the claims that azido presentation alone on the cell surface has an impact on DC biology should be presented in this context with appropriate controls. For example, the requirement for the azido group to be presented as a sialoside could/should be tested by evaluating other analogs used for metabolic glycan labeling such as azido-modified GalNAc or azido-modified fucose; these experiments would help unravel whether sialic acid per se is needed. Another way to address the role of sialic acid is to consider that sialylation is a regulator of galectin lattice interactions, a topic that has been studied by FRAP in the context of EGFR trafficking (e.g., doi: 10.1083/jcb.200611106 and doi: 10.18632/oncotarget.11582); note that the latter study modulates sialylation through ManNAc analog treatment conceptually similar to the experiments covered in this study. In this study an appropriate control analog would be Ac₄ManNAc, which increases flux into the sialic acid pathways to test if increased sialylation has an effect on the DC endpoints measured in this study (presumably Ac₄ManNAz both increases flux through the sialic acid pathway as well as introducing non-natural sialic acids onto the cell surface; this study does not address which of these two effects is responsible [or more highly responsible, they likely both have a role]). To follow up, if Ac₄ManNAc, which increases cell surface display of Neu5Ac (the predominant form of this sugar in humans), does not elicit the DC endpoints covered in this study, do other non-natural ManNAc analogs [I'm guessing that there are at least 20 of them reported] have this effect or is it unique to azido analogs?)

Response: We thank the reviewer for these comments. (i) We have now supplemented an experiment to compare the DC activation effect of Ac₄ManNAz and ManNAc. As shown in Fig. S3, Ac₄ManNAz and ManNAc resulted in a similar expression level of CD86 and MHCII, both were higher than the non-treatment group. These experiments demonstrated that the metabolic glycan labeling process, instead of the azido tag alone, is responsible for the improved activation of DCs. (ii) We also supplemented an experiment to compare the DC-activation effect of different types of azido-monosaccharides including Ac₄ManNAz, Ac₄GalNAz, and Ac₄GluNAz. All three azido-monosaccharides were able to upregulate the expression of CD86 and MHCII by DCs in a concentration-dependent manner, further substantiating the correlation between altered membrane expression of glycans and activation status of DCs.

We have now added Fig. S3, Fig. S4, and the following text “To understand whether the azido tag alone plays a role in DC stimulation, we compared the DC activation effect of Ac₄ManNAz and *N*-acetylmannosamine (ManNAc). Similar to Ac₄ManNAz, ManNAc was also able to upregulate the surface expression of CD86 and MHCII on DCs in comparison with untreated DCs (**Fig. S3a-d**). Compared to DCs treated with Ac₄ManNAz, DCs treated with ManNAc expressed similar levels of CD86 and MHCII (**Fig. S3a-d**), ruling out the impact of azido tag alone on the activation of DCs. We also expanded our analysis to include other non-natural sugars such as tetraacetyl-*N*-azidoacetylgalactosamine (Ac₄GalNAz) and tetraacetyl-*N*-azidoacetylglucosamine (Ac₄GluNAz). Both Ac₄GalNAz and Ac₄GluNAz treatment resulted in the upregulated expression of CD86 and MHCII on DCs, in comparison with untreated DCs (**Fig. S4a-b**)” in the revised manuscript.

Fig. S3. Azido tag itself does not further induce the activation of DCs. (a) Percentages of CD86⁺ DCs and (b) mean CD86 fluorescence intensity of DCs after 3-day incubation with Ac₄ManNAz or ManNAc. Also shown are (c) percentages of MHCII⁺ DCs and (d) mean MHCII fluorescence intensity of DCs after 3-day incubation with Ac₄ManNAz or ManNAc. All the numerical data are presented as mean ± SD (0.01 < **P* ≤ 0.05; ***P* ≤ 0.01; ****P* ≤ 0.001).

Fig. S4. DC activation effect of different types of azido-sugars. (a) Percentages of CD86⁺ DCs after 3-day incubation with different sugars. (b) Percentages of MHCII⁺ DCs after 3-day incubation with different sugars.

(3) Useful information to include is the “washout” rate of cell surface azido groups; i.e., what is the decay rate after removal of Ac₄ManAz? I suspect that it is substantial during the 16 hour incubations when tagging the cells with peptide or ILs. Note that this comment is not intended to be critical of the loss of cell surface labeling per se; rather the authors could make the point that the labeling, although transient of itself, sets a durable set of downstream responses in motion of (potential) therapeutic value).

Response: We thank the reviewer for this comment. We have supplemented an experiment to monitor the density of cell-surface azido groups over time after removing the azido-sugars in the culture media, and provided the data as **Fig. S12**. The density of cell-surface azido groups was well maintained over 72 h and started to significantly decrease after 96 h.

We have now added **Fig. S12** and the following text “It is noteworthy that azido groups expressed on the membrane of DCs can well retain for at least 72 h (**Fig. S12**), providing a durable time window for *in vivo* targeting of DBCO-bearing molecules” in the revised manuscript.

Fig. S12. Turn-over rate of cell-surface azido groups. BMDCs were pretreated with Ac₄ManNAz for three days and then transferred to fresh medium, followed by the detection of cell-surface azido groups using DBCO-Cy5 at different times.

(4) Another methodology-related issue is that no explanation for why P400 was used is given. For example, was it shown to be a better labeling reagent than Ac₄ManAz in previous studies (in the present study it didn't seem any better and at some point seemed to be dropped from the experiments with no clear explanation why that was done either)? Also, the way the concentrations are presented comparing Ac₄ManAz and P400 as given don't seem plausible (i.e., that both were used at 200 μ M or 400 μ M; does this mean that ~400-fold the molar equivalent of ManAz was delivered in the P400 experiments considering that each copy of P400 supplies 400 ManAz's? (I assume the dose was adjusted according to supply the same equivalents but this doesn't seem to be explained anywhere, it should be, and the figures need to be labeled to address this issue as well).

Response: We appreciate this comment. As Reviewer #3 also suggested, we have now deleted all the data and information related to P400 in the revised manuscript, which we agree makes the manuscript more coherent and concise.

(5) Detail of methods: The FRAP assay needs more detail; in particular, how many washing steps were done, e.g., to remove residual FBS and what were the exact media conditions used? This is important because any residual FBS could supply an exogenous source of galectins, which could affect the results (the role of galectin and sialic acids has been raised above)

Response: During the FRAP assay, we washed the cells for 4 times. We have now added “After washing for four times, cells in a culture dish were placed under a confocal microscope in a live cell imaging chamber” to the Method section.

(6) This point was also alluded to above, but the methods do not explicitly describe the experiments where the DCs were treated with analog for 72 h and then treated subsequently for 16 h (or other time points) with the SIINFEKL peptide (or whatever). Was Ac₄ManAz maintained in the system during these 16 hours? (I’m assuming not because although its inclusion would help maintain cell surface levels azido moieties, the free analog would compete for conjugate formation, preventing facile surface labeling [just curious, have experiments actually been done to see which effect would be greater?])

Response: Ac₄ManNAz was maintained during the incubation with the SIINFEKL peptide, as SIINFEKL incubation does not involve conjugation. Later when we performed the conjugation of DBCO-cytokines or DBCO-fluorophores, fresh media without azido-sugars were used. We have now added “... and then incubated with the SIINFEKL peptide or OVA protein of varying concentrations for 16 h in the presence of Ac₄ManNAz” to the Method section.

(7) Minor points: Throughout: is the term “membrane mobility” shorthand for “mobility of proteins in (the) membrane”? To me the authors are more accurately / precisely referring to “membrane fluidity” (i.e., the viscosity of the lipid bilayer and how this affects the movement of membrane components)

Response: Our FRAP study indicated the altered mobility of proteins on DC membrane, which is what we intended to express with ‘membrane mobility’. Membrane fluidity is also a great suggestion although it may somehow remind readers of lipid-related properties. We have now clarified ‘mobility of proteins on DC membrane’ in the main text.

(8) Page 2, last sentence of first paragraph: should “the benign safety profile” be “their benign safety profile”? Page 2, second paragraph: should “explored as DC source” be “explored as a DC source” or “explored as the DC source”?

Response: Thanks for catching these. We have corrected them in the revised manuscript.

(9) Page 2, second paragraph: does the word “could” (in “could undermine the resultant CTL”) imply that the rest of the sentence is just hypothetical? (for example, compare the meaning of the sentence if “often” or “are factors that” were used instead of could).

Response: We have replaced ‘could’ with ‘often’ in this sentence.

(10) Page 3, RESULTS section (and throughout): “Ac₄ManAz” is more commonly (and accurately) written as “Ac₄ManNAz” (i.e., adding the “N” before Az makes it explicit where the azido group occurs in the molecule).

Response: We have now replaced Ac₄ManAz with Ac₄ManNAz throughout the manuscript.

(11) Page 5: The sentence that includes “SIINFEKL, considering the well-established characterization methods” should include a reference to these methods Page 5, a few lines later: Would just “Consistent with” be better than “In consistence with”?

Response: We have now added references to this sentence. We have also revised “In consistence with” to “consistent with”.

(12) Page 12, “Cell lines and animals” section: Penicillin/Streptomycin do not need to be capitalized and CO₂ should be CO₂. Where was the FBS purchased from (perhaps this is listed in the Materials section but I’m not finding it)? Page 12, Synthesis of Ac₄Man(N)Az: The synthesis of this compound is “well known” and probably doesn’t need to be described in detail (if the journal has no space limitations, I suppose that inclusion of this information is fine) unless the authors use a revised/new synthetic route (did they?)

Response: Thanks for these detailed suggestions. We have now edited the sentences accordingly. We have also removed the synthetic procedure of Ac₄ManNAz and instead provided the reference for those procedures in the revised manuscript.

(13) Page 13, second paragraph: A more descriptive name for “Ac₃Man(N)AzOH” would be 3,4,6-OAc₃ManNAz (i.e., the locations of the acetyl groups would be specified).

Response: Since we have removed all P400-related information, this part has also been deleted in the revised manuscript.

(14) Page 16, first line (and near the end of the page): To be clear on the dose of “DC vaccines (1X10⁶ per mouse)”, does that mean 1X10⁶ cells per mouse?

Response: Yes, it is 1 x 10⁶ cells per mouse. We have updated it in the revised manuscript.

(15) Page 21, Figure 1 legend: (note that this point also arises in the main text). What does “targeted conjugation” mean? As the authors point out here in the figure legend, azido-sugars occur on glycoproteins and glycolipids, which is indeed correct, and furthermore, occur on many/most cell surface glycoproteins and glycolipids. From this perspective, the cell surface labeling of membrane components is rather ubiquitous and not really targeted per se. To me, replacing “targeted conjugation” with something like “cell surface conjugation” would be considerably more accurate. Also, considering that the authors mention the glycolipids can be modified with the azido-bearing sugars, can (should?) this be shown in their cartoons? (considering that lipid / membrane fluidity is affected, this could be directly relevant...)

Response: We actually intended to express ‘click chemistry-mediated targeting of cytokines to azido-tagged DCs’. To avoid confusion, we have revised ‘targeted conjugation’ into ‘conjugation’ throughout the manuscript. Regarding the metabolic tagging of lipids, many researchers in the field of metabolic glycan labeling are aware that some lipids will also be tagged. However, it is more challenging to characterize azido-tagged lipids than azido-tagged proteins. For this reason, most papers focus on showcasing the azido-tagged proteins. In this manuscript, we also want to focus on proteins, so if the Reviewers agree, we would like to keep the schematics as it is.

(16) Page 23, Figure 2 legend: “ate” should be “are” (typo) in (d); in (g) “after treated” should be “after treatment” or “after being treated”

Response: We have now corrected these typos.

(17) Page 25, Figure 3 legend: The term “Ac4” shown on the figure should be defined in the figure legend.

Response: We now have defined Ac4 in the figure legend.

Reviewer #2

Han et al., report the metabolic glycan labelling on dendritic cells improved the antigen presentation and antitumor immune response and demonstrate the application of the approach to target conjugate useful cytokines to improve the effector cell response. Earlier studies demonstrated the potential of a gel based in situ labelling approach to target the DCs and take the advantage of click chemistry to label cytokines of interest to improve the T cell expansion. Current study focus on the in vitro labelling and vaccination of antigen loaded dendritic cells as vaccine for cancer immunotherapy. Authors demonstrated the efficacy of the approach using in vivo models and 50% of the vaccinated animals shows a long-term survival. In general, the study is well conducted and there are some concerns as listed below.

Response: We thank the reviewer for the positive comments, and have addressed the questions accordingly below.

(1) The study shows an effective antitumor immune response and the underlying mechanism is not well defined. Authors claims the treatment with Ac4ManAz or P400 improved DC activation (Fig 1) based on the expression of CD86 and MHC II. It will be great to assess the expression of additional markers such as CD40, CD83 or CCR7 to confirm there is a really a DC activation.

Response: We have now analyzed the expression of additional markers including CD40 and CCR7, (Fig. 2e, Fig. 2f, Fig. S2b, and Fig. S2c). As expected, Ac4ManNAz-treated DCs showed an upregulated expression of CD40 and CCR7 in comparison with untreated DCs.

We have now added Fig. 2e-f and Fig. S2b-c, and added the following text “Ac₄ManNAz-treated DCs also expressed a significantly higher number of CD40 and CCR7, additional activation markers of DCs, than untreated DCs (Fig. 2e-f, Fig. S2b-c)” in the revised manuscript.

Fig. (2e) Mean CD40 FI and (S2b) percentage of CD40⁺ BMDCs after treatment with different concentrations of Ac₄ManNAz for three days and incubation with APC-conjugated anti-CD40. (2f) Mean CCR7 FI and (S2c) percentages of CCR7⁺ BMDCs after treatment with different concentrations of Ac₄ManNAz for three days and incubation with Alexa Fluor 700-conjugated anti-CCR7.

(2) The cytokine profile does not show any significant difference on IL6 or IL-12 to confirm the treatments activate the DCs (Fig S3). The observed change in CD86 and MHC may be due to the glycosylation of the molecules due to the glycan treatment and that may reduce the ubiquitination and turnover rate of these molecules. The same concept is applicable for the ICAM-1 and the observed stability may be due to the reduced turnover rate and lack of ubiquitination of the ICAM-1 molecule.

Response: We have now provided more evidence for the improved activation of Ac₄ManNAz-treated DCs including the upregulated CD40 and CCR7. Among the cells that were treated by Ac₄ManNAz, azide⁺ subpopulation clearly showed a much higher expression of CD86 and MHCII (Fig. 2g-i, Fig. S2d-f). Per the suggestion from another Reviewer, we have also performed transcriptome analysis of DCs, which showed that Ac₄ManNAz treatment caused radical and widespread transcriptional changes. We believe these data, together with many other data, well support the claim that Ac₄ManNAz treatment can induce the activation of DCs.

Figs 2g-i and S2d-f. Azide⁺ DC subpopulation shows significantly higher expression of CD86 and MHCII.

Fig. 2j. Altered gene expression profiles between Ac₄ManNAz-treated DCs and untreated DCs. DCs were treated with 200 μM Ac₄ManNAz or PBS for three days.

Fig. S6. Metabolic glycan labeling radically alters the gene expression of DCs. Normalized expression of (a) H2-K2 (MHCI), (b) H2-D1 (MHCI), (c) IL-15, (d) IL1α, (e) TNF-α, (f) CD86, and (g) Spi-C mRNAs, as determined by the transcriptome analysis.

(3) Authors claims that Ac₄ManAz treated DCs have advantage on antigen processing and presentation shows a better T cell activation and antigen presentation. The improved T cell activation can be directly correlated to the high MHC molecules on treated DCs, than antigen processing. If treated and untreated molecules have a similar level of MHC and there is difference in antigen presentation there can be an inherent difference in their antigen processing ability. It will be great to understand what is the advantage Ac₄ treated DCs compared to untreated DCs to elicit the antitumor immune response? Also the contribution of tissue resident DCs as well as other immune cells such as NK cells in the observed tumor regression. Nanoparticle or VLPs decorated with saccharides demonstrated an improved lymph node draining and improved the antigen specific T cell response. The Ac₄ treated DCs may have an improved lymph node draining and may be the tissue resident DCs may be capturing the antigens through cross-dressing and activate T cells. It is critical to evaluate the direct and indirect antigen presentation in this context to understand the how exactly the Ac₄ treated DCs improve the T cell response. A recent study

clearly demonstrated that the BMDCs are incapable of direct antigen presentation and testing the vaccination in a BATF3^{-/-} or IRF8^{32^{-/-}} mouse model will be critical to understand the contribution of lymph node resident DCs in T cell activation. The Ac₄ treated DCs may have an advantage to carry the antigen to the draining lymph node due to glycan treatment and results in the better presentation. The authors should address the underlying mechanism and it is important for translational settings where the endogenous cDC1s in cancer patients are generally compromised in their frequency and function.

Response: We appreciate these valuable comments from the Reviewer. For the comment that “BMDCs are incapable of direct antigen presentation”, we would like to clarify that the expression of MHCI-SIINFEKL on the membrane of BMDCs was detected via anti-MHCI-SIINFEKL (**Fig. 3a-b, 3g**). Others also reported the successful presentation of antigens such as SIINFEKL via MHCI complexes by BMDCs (Immunology 2006, 117, 78–88; Nature Immunology 2005, 6, 107–113; Nature Immunology 2003, 4, 1065–1073). Considering that Ac₄ManNAz-treated DCs were able to express a higher amount of MHCI-SIINFEKL complex than control DCs in the presence of the same concentration of SIINFEKL peptide or ovalbumin protein (ovalbumin needs additional degradation steps), it is cautiously safe to claim that Ac₄ManNAz-treated DCs exhibit improved antigen processing and presentation ability. Taking a step back, if Ac₄ManNAz treatment improves the expression of MHCI and then MHCI-Ag complexes, this should also be counted as an advantage for Ac₄ManNAz-treated DCs versus untreated DCs.

Regarding the effect of Ac₄ManNAz treatment on DCs, as we responded above, we now have more data to support the claim that Ac₄ManNAz treatment significantly improved the activation status of DCs: (1) DCs treated with Ac₄ManNAz express a significantly higher number of CD86, MHCII, CD40, and CCR7. (2) Among the cells that were treated by Ac₄ManNAz, azide⁺ subpopulation clearly showed a much higher expression of CD86 and MHCII (Fig. 2g-i, Fig. S2d-f). (3) Transcriptome analysis showed that Ac₄ManNAz-treated DCs exhibited radically changed gene expression in comparison with untreated DCs. We believe the enhanced activation status of DCs, as a result of Ac₄ManNAz treatment, contributes to the improved processing and presentation of antigens and further imparts the enhanced priming of antigen-specific CTLs.

To answer whether Ac₄ManNAz-treated DCs exhibit an enhanced lymph node draining property, we subcutaneously injected Calcein AM-stained DCs that were pretreated with Ac₄ManNAz or PBS. A negligible difference in the amount of Calcein AM⁺ DCs was detected in the draining lymph nodes between the DC-N₃ and DC groups (**Fig. S14d**), ruling out the possibility that Ac₄ManNAz-treated DCs show improved lymph node draining property. We have now added “It is noteworthy that a similar amount of calcein AM⁺ DCs were detected in the draining lymph nodes between Ac₄ManNAz and PBS groups (**Fig. S14d**), demonstrating the unaltered lymph node draining property of DCs after Ac₄ManNAz treatment” in the revised manuscript.

There is no doubt Ac₄ManNAz- and SIINFEKL-presenting DCs can prime SIINFEKL-specific T cells, as evidenced by our DC-OT1 co-culture studies and vaccination studies. Regarding whether Ac₄ManNAz-treated DCs can transfer antigens to resident DCs in the lymph nodes and then resident DCs prime SIINFEKL-specific CD8⁺ T cells, we do not envision a high efficiency or contribution from this process, if there is any.

Fig. S14. *In vivo* conjugation of DBCO-Cy5 to adoptively transferred, azido-labeled DCs. (a) Timeframe of *in vivo* targeting study. DCs were pretreated with Ac₄ManNAz or PBS for three days and stained with Calcein AM, prior to injection into C57BL/6 mice. DBCO-Cy5 was administered at 24 h, and lymph nodes were harvested for analysis at 30 h. (d) **Number of Calcein AM⁺ DCs in the draining lymph nodes.**

(4) *In vivo* targeting of IL-15 is an interesting approach and it will be great to show dot plots with other cells including *in situ* DCs to demonstrate that the DBCO binding is specific for injected DCs. It is also interesting to check the frequency of NK cells upon the IL-15 injection.

Response: We have supplemented the analysis of other cells including resident DCs (Fig. S14). As shown in Fig. S14c, DBCO-Cy5 showed minimal accumulation in resident DCs (calcein AM⁻ DCs), natural killer cells, T cells, and neutrophils. Also, the adoptive transfer of Ac₄ManNAz-pretreated DCs or untreated DCs did not result in any noticeable changes in the number of immune cells such as natural killer cells in the draining lymph nodes (Fig. S14e).

We have now added Fig. S14 and the following text “To study the *in vivo* targeting of azido-tagged DCs, BMDCs pretreated with Ac₄ManNAz or PBS were stained with calcein AM and injected into C57BL/6 mice, followed by the injection of DBCO-Cy5 at 24 h. At 6 h post injection of DBCO-

Cy5, lymph nodes were harvested for flow cytometry analysis (**Fig. 6a, Fig. S14a-b**). Compared to DCs without azido-sugar pretreatment, Ac₄ManNAz-pretreated DCs exhibited higher Cy5 fluorescence signal (**Fig. 6b-c**), indicating the successful conjugation of DBCO-Cy5 onto azido-labeled DCs *in vivo*. In contrast, endogenous DCs in the draining lymph nodes, i.e. calcein AM⁺ DCs, showed minimal Cy5 signal (**Fig. S14c**). T cells, natural killer cells, and neutrophils in the draining lymph nodes also showed minimal Cy5 accumulation (**Fig. S14c**). These data demonstrated the specific conjugation of DBCO-Cy5 onto adoptively transferred, azido-tagged DCs. It is noteworthy that a similar amount of calcein AM⁺ DCs between Ac₄ManNAz and PBS pretreatment groups were detected in the draining lymph nodes (**Fig. S14d**), demonstrating the unaltered lymph node draining property of DCs with or without Ac₄ManNAz pretreatment. The adoptive transfer of Ac₄ManNAz-pretreated DCs or untreated DCs did not result in any noticeable changes in the number of immune cells such as natural killer cells in the draining lymph nodes (**Fig. S14e**)” in the revised manuscript.

Fig. S14. *In vivo* conjugation of DBCO-Cy5 to adoptively transferred, azido-labeled DCs. (a) Timeframe of *in vivo* targeting study. DCs were pretreated with Ac₄ManNAz or PBS for three days and stained with Calcein AM, prior to injection into C57BL/6 mice. DBCO-Cy5 was administered at 24 h, and lymph nodes were harvested for analysis at 30 h. (b) Representative gating strategy for analyzing Cy5⁺Calcein AM⁺ cells. (c) Mean Cy5 fluorescence intensity of different immune cells in lymph nodes. (d) Number of Calcein AM⁺ DCs in the draining lymph nodes. (e) Number of natural killer cells in the draining lymph nodes. All the numerical data are presented as mean ± SD (0.01 < *P ≤ 0.05; **P ≤ 0.01; ***P ≤ 0.001).

(5) Prophylactic tumor study (Figure 4) : the Schema shows the tumor cells are implanted on day 21 and Figure 4d. shows a tumor growth on day 10.

Response: We thank the reviewer for this comment. We actually counted the date of tumor cell inoculation as day 0 for tumor monitoring. We have now clarified this in the figure legend.

Reviewer #3:

In this manuscript, Han et al reported a metabolic labeling approach for the targeted delivery of T-cell-stimulating cytokines to the adoptively transferred DC vaccine. Interestingly, they show that metabolic glycan labelling itself enhances DC maturation, antigen cross-presentation. Combination of labeled DC vaccine and *in vivo* administered cytokines improve the anticancer efficacy. This chemical-conjugation-based approach is interesting and provides a new way to target transferred DCs *in vivo*. The *in vitro* and *in vivo* experiments were well designed to demonstrate the targeting effects. Most interpretations were reasonable and conclusions were supported by experimental evidence. There are several issues that should be addressed to improve the current manuscript.

(1) It is interesting that the authors found that metabolic glycan labelling itself can enhance maturation of DCs. However, this phenomenon needs to be further confirmed with additional experiments: 1) controls groups including sugar alone (no azide group), polymer alone (no sugar side chains), and sugar-azide monomer should be added to investigate which part indeed contributes to the observed effect; 2) Fig 2g-i, it is possible that the fluorescence of Cy5 bleeds to other channels. To exclude that possibility, the author may use the same color (as the one for CD86) to label a non-changed marker (e.g., CD45) and show no difference between Cy5⁺ and Cy5⁻ groups; 3) other maturation markers including CD80, MHC-II should be assessed (ideally with different colors, for the reason of 2)

Response: We thank the reviewer for this comment.

1) Per the suggestion from multiple Reviewers, we have now deleted all the data related to P400. We have compared the DC activation effect of Ac₄ManAz and ManNAc (Fig. S3). It turned out that ManNAc treatment also upregulated the expression of CD86 and MHCII by DCs.

2) We used FITC for the CD86 or MHCII, so the spill over to Cy5 should be minimal. Compensation was also applied during the flow cytometry analysis.

3) We have provided MHC-II data as Fig. 2 I and Fig. S2e-f (FITC-conjugated anti-MHCII was used). Consistent with the CD86 data, Cy5⁺ subpopulation showed a significantly higher expression level of MHC-II than Cy5⁻ subpopulation.

Fig. S3. Azido tag itself does not further induce the activation of DCs. (a) Percentages of CD86⁺ DCs and (b) mean CD86 fluorescence intensity of DCs after 3-day incubation with Ac₄ManNAz or ManNAc. Also shown are (c) percentages of MHCII⁺ DCs and (d) mean MHCII fluorescence intensity of DCs after 3-day incubation with Ac₄ManNAz or ManNAc. All the numerical data are presented as mean ± SD (0.01 < *P ≤ 0.05; **P ≤ 0.01; ***P ≤ 0.001).

Figs 2i and S2e-f. Azide⁺ DC subpopulations show significantly higher expression of MHCII.

(2) The mechanism underlying the enhanced maturation/antigen presentation of DC by metabolic glycan labelling was not clear. The authors showed the results of photobleaching for a measurement of membrane fluidity. However, the impact on membrane fluidity seems minimum

(Fig 2j, k; Figure S4), and whether it is a causation (rather than a correlation) remains to be determined. This mechanistic study is weak and better removed.

Response: We thank the Reviewer for this comment. While the mechanism underlying the enhanced activation and antigen presentation of DCs by metabolic glycan labeling is not entirely clear, the membrane fluidity data together with the altered gene expression profiles could be valuable information for further elucidation of the underlying mechanisms. Considering that other Reviewers especially Review #1 really likes this study, we would like to keep the data in the revised manuscript.

(3) Fig 2: it is not clear why the authors investigated P400 here. If I understand correctly, for all the rest of the results (Fig 3-6), only Ac4ManAz was used for metabolic labeling of DCs. If so, the data of P400 are not relevant.

Response: Per the suggestions from multiple Reviewers, we have now deleted all the information and data related to P400 in the revised manuscript.

(4) Fig 3: the data for SIIFEKL or OVA protein are similar, please keep one and move the other to SI. Please use consistent colors to label groups w or w/o Ac4. Why different concentrations of OVA were used in j and k?

Response: The processing of OVA by DCs is often deemed more challenging than SIINFEKL peptide, so showing data for both might enable the readers to have a better understanding of the impact of metabolic glycan labeling on the antigen presentation ability of DCs.

(5) From the OT-1 CD8 T cell proliferation data shown in Fig 3, it seems that metabolic glycan labelling enhances antigen cross-presentation. What is the possible mechanism, endosomal escape of the antigens? Please provide evidence for possible mechanisms.

Response: The enhanced OT-1 proliferation can be explained by the higher expression level of MHCII-SIINFEKL and co-stimulatory signals (CD86 and CD40) by Ac4ManNAz-treated DCs in comparison with control DCs (Fig. 2d-i, Fig. 3a-b).

(6) Fig 5f, g: what if same amount of IL-15 (or IL-2) was added directly to the co-culture rather than pretreatment of DCs with cytokines?

Response: We thank the reviewer for this comment. We actually included the continuous cytokine treatment when performing the co-culture studies and have now provided the comparison between cytokine conjugation group and continuous cytokine group in Fig. S11c-f. At 5 ng/mL, the IL-15 conjugation group showed negligible differences in OT-1 proliferation compared to the continuous IL-15 group. In contrast, the IL-2 conjugation group showed improved OT-1 proliferation compared to the continuous IL-2 group. At 25 ng/mL, the IL-15 conjugation group resulted in slightly higher OT-1 proliferation than the continuous IL-15 group. IL-2 conjugation group showed negligible differences in OT-1 proliferation compared to the continuous IL-2 group.

We have added Fig. S11c-f and the following text “It is noteworthy that the conjugation of IL-15 or IL-2 onto DCs resulted in comparable or higher expansion rates of OT-1 cells in comparison with the co-culture systems where IL-15 or IL-2 was present throughout the culture period (Fig. S11c-f), further demonstrating the superior T cell priming capability of IL-15/IL-2 conjugated DCs” in the revised manuscript.

Fig S11. (c,e) Proliferation index of OT-1 cells after 3-day coculture with IL-15-conjugated DCs or control DCs + continuous IL-15 incubation. (d,f) Proliferation index of OT-1 cells after 3-day coculture with IL-2-conjugated DCs or control DCs + continuous IL-2 incubation.

(7) The chemical approach for targeting cytokines to transferred DCs is certainly interesting and new. However, there are several issues: 1) how long can the surface-bound cytokine stay? DCs also express the common cytokine receptor gamma chain including IL-2 receptor (IL-2R) (<https://academic.oup.com/intimm/article/10/2/237/674045>). Once bind to the cytokine receptor, the surface cytokines may be quickly internalized by DCs and therefore have no effects on the T cells engaged; 2) the cytokines presented on DC surface still have the potential to trigger activation of tumor-antigen non-specific T cells. How much gain in safety as compared to administration of equivalent cytokine remains to be determined; 3) when DCs are properly activated, they naturally secrete cytokine as signal 3 to boost T cells engaged. Additional surface-bound cytokine may have minimum further improvement in this action.

Response: We thank the reviewer for these comments. (i) Considering the tiny amount of cytokines we can use to conjugate to DCs, we examined the conjugation and membrane retention of DBCO-Biotin (via detection with FITC-avidin). As shown in Fig. S13, the conjugated biotin was well retained on DC membrane for 72 hours, with a significant drop in density from 96 hours. It is true that the conjugated IL-2, if bound to IL-2 receptor, could be rapidly internalized. However, the covalently conjugated IL-2 might not be in a good conformational state to efficiently bind with IL-2 receptors. In view of the improved OT-1 proliferation in the conjugation group, it is likely that the conjugated IL-2 and IL-15 could retain on DC membranes for a certain amount of time. (ii) To examine how cytokines presented on DC surface can trigger activation of tumor-antigen non-specific T cells, we supplemented an experiment in which we co-cultured cytokine-conjugated DCs with both OT-1 and non-specific CD8⁺ T cells. As shown in Fig 5 h-j, the proliferation index of OT-1 cells was significantly higher than non-specific CD8⁺ T cells. It is noteworthy that the

proliferation index of non-significant T cells did not show any difference between cytokine-conjugated DCs and control DCs. These data demonstrated that IL-2/IL-15-conjugated DCs can preferentially proliferate antigen-specific CD8⁺ T cells. (iii) In T cell priming processes, it is common to add additional T cell-activating signals such as IL-2 and IL-15 to further improve the priming of T cells. By conjugating IL-2 or IL-15 to antigen-presenting DCs, we were able to improve the generation of antigen-specific CD8⁺ T cells in vitro and in vivo. However, we agree that surface-bound cytokines may contribute to the stimulation of interacting T cells for only a certain duration.

We have now added the following text “To examine the membrane retention time of covalently conjugated molecules, BMDCs were pretreated with Ac₄ManNAz for three days and incubated with DBCO-biotin, followed by the detection of surface biotin groups via FITC-avidin at different times. It turns out that ~70% of covalently conjugated biotin could retain on DC membrane for 72 h (Fig. S13)” “To assess the potential off-target T cell activating effect of cytokine-conjugated DCs, we also co-cultured IL-15 conjugated DCs with both CFSE-stained OT-1 cells and Violet Stain-stained non-specific CD8⁺ T cells for three days. As a result, OT-1 cells showed a significantly higher proliferation rate than non-specific CD8⁺ T cells (Fig. 5h-j). Compared to non-specific CD8⁺ T cells that were co-cultured with DCs, cells cocultured with IL-15-conjugated DCs failed to exhibit improved proliferation (Fig. 5h-j)” in the revised manuscript.

Fig. S13. Membrane retention of molecules conjugated onto DCs. BMDCs were pretreated with Ac₄ManNAz for three days and then incubated with DBCO-Biotin for 30 min. DCs were then transferred to fresh media and cell-surface biotin was detected by FITC-avidin at different times.

Fig. 5. (h-j) IL-15 conjugated DCs or DCs were pulsed with SIINFEKL for 16 h and then co-cultured with CFSE-stained OT-1 cells and Violet-stained non-specific CD8⁺ T cells for three days. (h-i) CFSE or Violet stain histogram of OT-1 cells and non-specific CD8⁺ T cells. (j) Proliferation index of OT-1 and non-specific CD8⁺ T cells.

(8) Fig 6d-f: a better experiment to do is to transfer of DC vaccine together with naive OT-1 T cells in vivo, and compare expansion of OT-1 vs. endogenous T cells (as antigen non-specific) in mice receiving cytokines or DBCO-cytokines. DBCO-cytokines are expected to boost more OT-1 T cells but less endogenous T cells.

Response: We thank the reviewer for this comment. As we responded above (comment #7), we supplemented an experiment in which we co-cultured cytokine-conjugated DCs with both OT-1 and non-specific CD8⁺ T cells. As shown in Fig 5 h-j, the proliferation index of OT1 cells was significantly higher than non-specific CD8⁺ T cells. It is noteworthy that the proliferation index of non-significant T cells did not show any difference between cytokine-conjugated DCs and control DCs. These data demonstrated that IL-2/IL-15-conjugated DCs can preferentially proliferate antigen-specific CD8⁺ T cells.

Fig. 5. (h-j) IL-15 conjugated DCs or DCs were pulsed with SIINFEKL for 16 h and then co-cultured with CFSE-stained OT-1 cells and Violet-stained non-specific CD8⁺ T cells for three days. (h-i) CFSE or Violet stain histogram of OT-1 cells and non-specific CD8⁺ T cells. (j) Proliferation index of OT-1 and non-specific CD8⁺ T cells.

(9) Fig 6g,h: tumor growth was almost identical but why the survival curves differ so much?

Response: The difference started to become significant from day 60 (Fig. 6h), while Fig. 6g gives the short-term comparison of tumor growth kinetics. This is not uncommon for vaccine studies.

Minor issues:

(10) For the same flow cytometry data, showing either frequency, counts, or MFI is enough. No need to duplicate the same data with two different forms of data presentation (Fig 1, 2)

Response: We appreciate the comment from the Reviewer. As the Reviewer may understand, many readers including Reviewers prefer to see a full set of analyses (both percentages and MFI).

We have now provided as much data as the Reviewers requested, and can work with the editor to reduce the amount of figures later if needed.

(11) The endpoints (tumor size, health condition, etc) for mouse tumor models studies should be stated in the method part.

Response: We have now added the endpoint information “Mice were euthanized when the largest diameter of tumors reaches 20 mm or mice became moribund” in the revised manuscript.

REVIEWER COMMENTS

Reviewer #1 (Remarks to the Author):

The revised manuscript adequately addresses my concerns from the first review process (indeed, the authors were highly attentive to my concerns, including additions experiments [albeit that were also requested by the other reviewers as well])

One minor suggestion (which does not require re-review by me) is for the authors to remove hyperbolic language such as "radical" or "radically" in relation to the results of the newly added genomic/transcript profiling data. In my opinion, the number of genes involved (~10 to 15% of the human genome) is more like the number that would be expected to change in during virtually any cellular perturbation; in other words, I find this result for the metabolically-glycoengineered cells to be "expected" not "radical"

Reviewer #3 (Remarks to the Author):

The authors have well addressed most of the previous comments. Some additional points to be addressed before publication (all related to the previous comments):

previous comment #2. The mechanism underlying the enhanced maturation/antigen presentation of DC by metabolic glycan labelling was not clear. The authors showed the results of photobleaching for a measurement of membrane fluidity. However, the impact on membrane fluidity seems minimum (Fig 2j, k; Figure S4), and whether it is a causation (rather than a correlation) remains to be determined. This mechanistic study is weak and better removed.

Still, the authors show no further studies to link the membrane fluidity and antigen presentation, which is not obvious. At least, the authors should make it explicit in the text that at this point, whether membrane fluidity is a causation of enhanced DC activation (rather than a correlation) remains to be determined in future studies.

previous comment #8. Fig 6d-f: a better experiment to do is to transfer of DC vaccine together with naive OT-1 T cells in vivo, and compare expansion of OT-1 vs. endogenous T cells (as antigen non-specific) in mice receiving cytokines or DBCO-cytokines. DBCO-cytokines are expected to boost more OT-1 T cells but less endogenous T cells.

The author did not perform the suggested in vivo experiment. However, the new Fig 5 h-j, as an in vitro experiment, to some extent, address the concerns.

previous comment #9. Fig 6g,h: tumor growth was almost identical but why the survival curves differ so much?

In Fig 6g, if the difference shows at day 60, please show the continued tumor growth curve of the two groups (until at least day 60, with the first mouse sacrificed in the group), DC-N3+DBCO-IL15 and DC-N3+IL15.

Point-by-point reply to reviewers comments

(All responses were colored in blue, and all changes in the manuscript were highlighted in yellow)

Reviewer #1

The revised manuscript adequately addresses my concerns from the first review process (indeed, the authors were highly attentive to my concerns, including additions experiments [albeit that were also requested by the other reviewers as well])

Response: We thank the reviewer for the positive comments.

(1) One minor suggestion (which does not require re-review by me) is for the authors to remove hyperbolic language such as "radical" or "radically" in relation to the results of the newly added genomic/transcript profiling data. In my opinion, the number of genes involved (~10 to 15% of the human genome) is more like the number that would be expected to change in during virtually any cellular perturbation; in other words, I find this result for the metabolically-glycoengineered cells to be "expected" not "radical"

Response: We have now deleted the term “radical” in the revised manuscript.

Reviewer #3

The authors have well addressed most of the previous comments. Some additional points to be addressed before publication (all related to the previous comments):

(1) previous comment #2. The mechanism underlying the enhanced maturation/antigen presentation of DC by metabolic glycan labelling was not clear. The authors showed the results of photobleaching for a measurement of membrane fluidity. However, the impact on membrane fluidity seems minimum (Fig 2j, k; Figure S4), and whether it is a causation (rather than a correlation) remains to be determined. This mechanistic study is weak and better removed. Still, the authors show no further studies to link the membrane fluidity and antigen presentation, which is not obvious. At least, the authors should make it explicit in the text that at this point, whether membrane fluidity is a causation of enhanced DC activation (rather than a correlation) remains to be determined in future studies.

Response: We thank the Reviewer for this comment. Although the link between the membrane fluidity and antigen presentation is not entirely clear, we believe the altered membrane fluidity together with the altered gene expression profiles could be valuable information for further elucidation of the underlying mechanisms. We have now toned down the mechanistic claims, removed ‘causal relationship’ wordings, and added “While the altered membrane fluidity could be related to the enhanced activation status of DCs, future efforts are needed to fully elucidate whether there is a causal relationship between them” in the revised main text.

(2) previous comment #8. Fig 6d-f: a better experiment to do is to transfer of DC vaccine together with naive OT-1 T cells in vivo, and compare expansion of OT-1 vs. endogenous T cells (as antigen non-specific) in mice receiving cytokines or DBCO-cytokines. DBCO-cytokines are expected to

boost more OT-1 T cells but less endogenous T cells. The author did not perform the suggested in vivo experiment. However, the new Fig 5 h-j, as an in vitro experiment, to some extent, address the concerns.

Response: We appreciate this comment from the Reviewer. We thought of performing the experiment suggested by the Reviewer, but a big concern is the low engraftment of the injected OT-1 cells in tissues including lymph nodes. It is hard to guarantee that the injected OT1 cells and endogenous T cells have equal chances to encounter the antigen-presenting DCs, which makes it hard to draw any meaningful conclusion. Therefore, instead, we supplemented the in vitro co-culture study where OT1 cells, naïve CD8⁺ T cells, and DCs are co-cultured to ensure that OT1 cells and naïve CD8⁺ T cells have equal chances to encounter DCs, and the data were presented as Fig. 5h-j.

(3) previous comment #9. Fig 6g,h: tumor growth was almost identical but why the survival curves differ so much? In Fig 6g, if the difference shows at day 60, please show the continued tumor growth curve of the two groups (until at least day 60, with the first mouse sacrificed in the group), DC-N3+DBCO-IL15 and DC-N3+IL15.

Response: We thank the Reviewer for this comment. For the DC-N₃ + DBCO-IL15 group, we had several tumor-free mice in the end, which is the reason why the survival curves differ more than the tumor growth curves. We have now updated the tumor growth curve (Fig. 6g) to include data points until day 62. As we can see, the average tumor size for the DC-N₃ + DBCO-IL15 group remained at a similar value after day 40 (several mice were tumor-free and several reached the endpoint).

Fig. 6g. Average E.G7-OVA tumor volume of each group over the course of prophylactic tumor study.